# Activate and Adapt:
# A Two-Stage Framework for Open-Set Model Adaptation

**Xiasi Wang**[1]*, **Jiaqi Lin**[2], **Chaoqi Chen**[3]†, **Luyao Tang**[4], **Yi Huang**[5], **Chengsen Wang**[6],
**Lei Ye**[7], **Yuan Yao**[1]

[1]The Hong Kong University of Science and Technology    [2]Tsinghua University
[3]Shenzhen University    [4]The University of Hong Kong    [5] University of Chinese Academy of Sciences
[6] Beijing University of Posts and Telecommunications    [7] Huawei

**Reviewed on OpenReview:** `https://openreview.net/forum?id=2AWbwSpET9`

## Abstract

The ability to generalize to new environments is critical for deep neural networks. Most existing works presume that the training and test data share an identical label set, overlooking the potential presence of new classes in test data. In this paper, we tackle a practical and challenging problem: Open-Set Model Adaptation (OSMA). OSMA aims to train a model on the source domain, which contains only known class data, and then adapt the trained model to the distribution-shifted target domain to classify known class data while identifying new class data. In this context, we face two challenges: (1) enabling the model to recognize new classes using only the known class data from the source domain during training, and (2) adapting the source-trained model to the target domain that contains new class data. To address these challenges, we propose a novel and universal two-stage framework named `Activate and Adapt (ADA)`. In the training stage, we extract potential new class information hidden within the rich semantics of the source domain data to enable the model to identify new class data. Additionally, to retain source domain information while preserving data privacy, we condense the source domain data into a small dataset, facilitating the subsequent adaptation phase. In the test stage, we adaptively adjust the source-trained model to the target domain with new classes by infusing the style of target data into the condensed dataset, and decoupling domain alignment for known and new classes. Experiments across three standard benchmarks demonstrate that `ADA` surpasses previous methods in both online and offline settings.

## 1 Introduction

Despite the remarkable success of machine learning in various fields, generalizing to new environments remains a significant challenge (Wang et al., 2022a). For instance, a model trained on urban environment data to identify cars may perform sub-optimally in the rural environment due to the distribution shift problem. Recently, considerable efforts have been dedicated to Domain Generalization (DG) (Zhou et al., 2022), which aims to enhance model performance on previously unseen domains.

Current approaches within the realm of DG include learning representations that are invariant to domain changes (Lv et al., 2022; Lu et al., 2022; Chen et al., 2023d; Wong et al., 2024), expanding data distributions through data augmentation and synthesis (Huang et al., 2021; Su et al., 2023; Bose et al., 2023; Zhu et al., 2023), and employing meta-learning to expose models to various shifts (Wang et al., 2023b; Chen et al., 2023c), among others. Despite these advances, most methods operate under the assumption that the source and target domains share an identical label space, which limits their applicability to the closed-set scenario.

---

  * Email to: xwangfy@connect.ust.hk
  † Corresponding author

A more practical and realistic concern is the existence of a broader label space in the target domain. For instance, a model trained with previous DG methods may mitigate the effects of distribution shift between urban and rural environments. However, when the model encounters objects specific to rural areas, such as livestock, it may struggle to recognize them due to their absence in the training data. This shortcoming could potentially lead to significant safety risks, compromising the reliability and safety of machine learning systems in diverse environments.

To enhance the adaptability of DNNs in an open-world context, we investigate a more practical and challenging problem: Open-Set Model Adaptation (OSMA). The goal of OSMA is to train a model on source domain data containing only known class data, and then adapt the model to perform effectively in the target domain, which is characterized by distribution shift and open class challenges. Specifically, the model needs to classify known class data while also identifying new class data in the target domain. In particular, during the adaptation phase, while direct access to original source data is restricted, it is allowed to utilize information extracted from the source data in a controlled manner (in our work, we condense the source data into a small dataset). We provide a comparison of OSMA with other settings in Table 1.

The critical challenges of OSMA lie in two aspects. Firstly, we need to equip the model with the ability to identify new (unknown) class data during the training stage, using only known class data from the source domain. Secondly, we need to adapt the source-trained model to the target domain for safe model deployment, ensuring its capability of classifying known class data while also identifying unknown class data.

To this end, we propose a novel and universal two-stage framework named `Activate and Adapt (ADA)`, which facilitates model adaptation under the dual challenges of distribution shift and open class. Our approach comprises three key components. Firstly, to address the first challenge, we propose *Unknown Activation*, which extracts the unknown class information hidden within the rich semantics of the known class data from source domain. This module enables the model to identify unknown class data during the training stage. Secondly, to harness the source domain data information during the adaptation phase while safeguarding data privacy, we introduce the *Source Condensation* module to condense the source data into a small dataset. Thirdly, we propose the *Open Model Adaptation* stage, which adapts the source-trained model on the target domain at test time. This is achieved by injecting style information from the target domain data into the condensed data to provide supervisory guidance, and inducing a decoupled domain-aligned adjustment for the cross-domain recognition of both known and unknown classes.

The contributions of our work are summarized as follows:

- We explore the Open-Set Model Adaptation problem and propose a two-stage framework, `Activate and Adapt`, to tackle the distribution shift and open class challenges in test environments.

- We leverage new class information hidden within the rich semantics of known class data to enable the model to identify new class data. Additionally, we introduce an innovative use of dataset condensation to preserve source domain information, facilitating the adaptation phase.

- Extensive experiments on three widely used standard benchmarks demonstrate that `ADA` outperforms existing methods in both online and offline settings.

## 2 Related Work

**Domain Generalization (DG)**. DG aims at enhancing model performance on unseen target domains. Existing DG methods can be categorized into different categories, such as data augmentation and generation (Huang et al., 2021; Su et al., 2023; Bose et al., 2023; Zhu et al., 2023), representation learning (Wang et al., 2022c; Lu et al., 2022; Wong et al., 2024), and meta-learning (Wang et al., 2023b; Chen et al., 2023c). Some works have considered the more realistic open-set DG (OSDG) scenario, where unknown classes appear in target domain. For example, CODA (Chen et al., 2023a) generates virtual unknown samples via manifold mixup (Verma et al., 2019). OneRing (Yang et al., 2022c) argues that any category not matching the ground truth should be regarded as the unknown class. However, the "adaptivity gap" (Dubey et al., 2021) persists when the source-trained model is directly deployed on the target domain without further adaptation.

Table 1: Comparison of different problem settings. $(x^s, y^s)$ is labeled source domain data and $x^t$ is unlabeled target domain data. OSMA gets extracted source domain data information $(\tilde{x}, \tilde{y})$ during training and utilizes it in the test stage. "Fine-tune" means updating the source-trained model parameters in the test phase; "Online" means the model adapts and predicts after a single pass over each data while "Offline" means the model adapts on the full test set before making predictions.

| Problem Settings | Training | | Test | | | | |
|---|---|---|---|---|---|---|---|
| | Training Data | Training Loss | Fine-tune | Adapt Loss | Open-set | Online | Offline |
| Open-Set DA | $x^s, y^s, x^t$ | $\mathcal{L}(x^s, y^s) + \mathcal{L}(x^s, x^t)$ | ✗ | – | ✓ | ✗ | ✗ |
| Source-Free DA | $x^t$ | $\mathcal{L}(x^t)$ | ✗ | – | ✗ | ✗ | ✓ |
| Open-Set DG | $x^s, y^s$ | $\mathcal{L}(x^s, y^s)$ | ✗ | – | ✓ | ✗ | ✗ |
| Test-Time Adaptation | $x^s, y^s$ | $\mathcal{L}(x^s, y^s)$ | ✓ | $\mathcal{L}(x^t)$ | ✗ | ✓ | ✗ |
| **OSMA** | $x^s, y^s$ | $\mathcal{L}(x^s, y^s)$ | ✓ | $\mathcal{L}(x^t, \tilde{x}, \tilde{y})$ | ✓ | ✓ | ✓ |

**Domain Adaptation (DA)**. The key difference between DG and DA is that DA has access to unlabeled target data for training while DG does not. Source free domain adaptation (SFDA) (Xia et al., 2021; Liang et al., 2020; Yang et al., 2022b; Wang et al., 2024) has been proposed to address the issue of source data unavailability when deploying the source-trained model to the target domain. For instance, SHOT (Liang et al., 2020) aligns cross-domain features using self-supervised pseudo labeling, and AaD (Yang et al., 2022b) employs prediction consistency and inconsistency to refine feature representations. Some works further study the label-set shift setting. For example, GLC (Qu et al., 2023) adopts clustering and pseudo-labelling techniques to identify known and unknown class data under category-shifts. LEAD (Qu et al., 2024) decouples features to identify target-private data. Inheritable model (Kundu et al., 2020) proposes the vendor-client paradigm to solve the open-set problem.

**Test Time Adaptation (TTA)**. TTA is an emerging paradigm that enhances model generalizability through unsupervised fine-tuning in an online manner (Wang et al., 2020; Sun et al., 2020; Chen et al., 2022; Gan et al., 2023; Wang et al., 2023a; Chakrabarty et al., 2023; Wu et al., 2024; Su et al., 2024). For example, Tent (Wang et al., 2020) proposes entropy minimization to improve model confidence. TTT (Sun et al., 2020) introduces auxiliary self-supervised tasks to fine-tune the model. A few works explore the open-set setting in TTA. For instance, ART (Chen et al., 2023b) uses cross-domain nearest neighbor and class prototype information to detect open-class data, and OSTTA (Lee et al., 2023) selects samples with higher confidence values to conduct entropy minimization.

**Comparison of Different Settings.** The comparison of OSMA and other settings is shown in Table 1. OSMA differs from OSDA and SFDA in that it does not access target domain data during training. Compared to OSDG, which does not adaptively adjust the source-trained model at test time, and TTA, which adapts the model to the target data only in an online manner, OSMA leverages extracted source domain information to adapt the source-trained model on target data, and it is applicable in both online and offline settings.

## 3  Method

**Design Overview.** Our proposed `Activate and Adapt (ADA)` consists of two stages. In the first stage, *Unknown Activation and Source Condensation (UASC)*, we equip the model with the ability to identify unknown class data using only known class data from the source domain for training. We also condense the source data to a small dataset to preserve information. In the second stage, *Open Model Adaptation (OMA)*, we utilize the condensed dataset along with target domain data to jointly adapt the model and make final predictions for the target domain data. The illustration of `ADA` is presented in Figure 1.

**Problem Setup.** We define the OSMA problem formally. In the training stage, we have $N_s$ labeled data from source domain $\mathcal{D}_s = \{(x_i^s, y_i^s)\}_{i=1}^{N_s}$. Our task is to classify $N_t$ unlabeled data from target domain $\mathcal{D}_t = \{x_i^t\}_{i=1}^{N_t}$. We denote the label sets of $\mathcal{D}_s$ and $\mathcal{D}_t$ as $\mathcal{C}_s$ and $\mathcal{C}_t$ respectively. The target domain encompasses a wider class scope than the source domain, i.e., $\mathcal{C}_s \subset \mathcal{C}_t$, $\mathcal{C}_t \setminus \mathcal{C}_s = \mathcal{C}_t^u$. We refer to $\mathcal{C}_s$ as known classes and $\mathcal{C}_t^u$ as unknown classes. In our setting, we take the unknown classes as a whole. Our goal is to

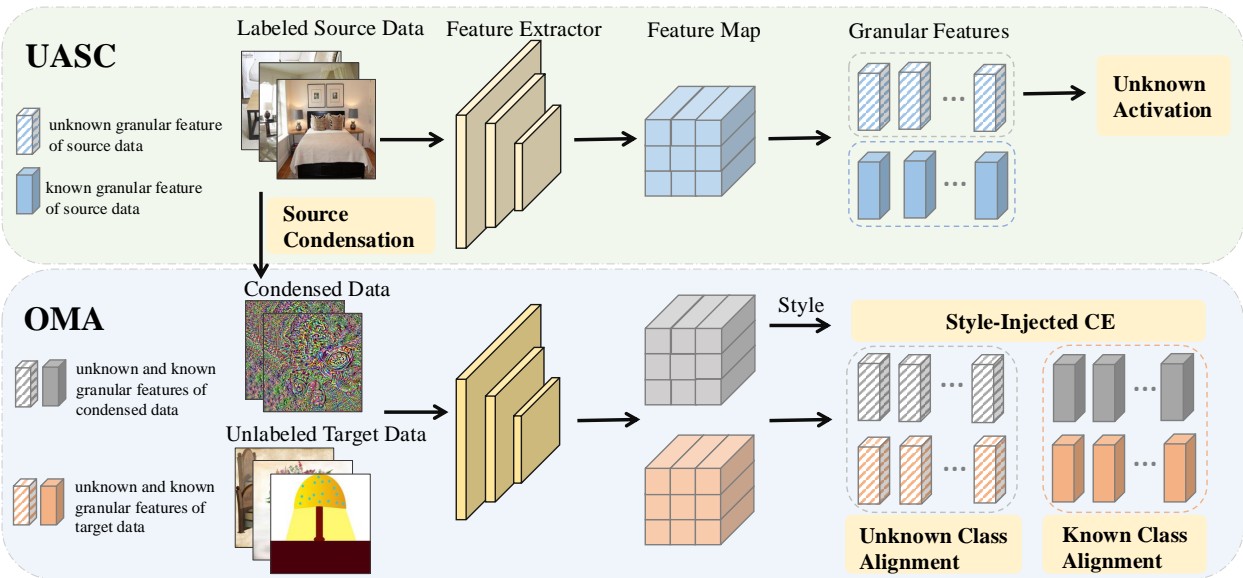

Figure 1: Illustration of `ADA`. `ADA` comprises two stages: *Unknown Activation Source Condensation (UASC)* and *Open Model Adaptation (OMA)*. In *UASC*, we decompose the feature maps of source domain data to granular features, and activate the classification logit for the unknown class dimension using the classified unknown class granular features. We additionally condense a small dataset to preserve source domain information. In *OMA*, we inject the target domain style information to the condensed data for supervisory guidance, and align cross-domain knowledge for both known and unknown classes.

train a model on $\mathcal{D}_s$ and then adapt the model on $\mathcal{D}_t$ to classify the data in $\mathcal{D}_t$ to $|\mathcal{C}_s| + 1$ classes, where the $(|\mathcal{C}_s| + 1)$-th dimension means the unknown class. Note that the original source domain data $\mathcal{D}_s$ is no longer available in the adaptation phase, while a lightweight proxy of the source data is allowed to be used.

## 3.1 Unknown Activation and Source Condensation

### 3.1.1 Unknown Activation

**Unknown Extraction.** DNNs trained with standard cross-entropy loss on known class data fail to identify unknown classes (Chen et al., 2023b). However, it is overlooked that images contain rich semantics (Zhou et al., 2016; Li et al., 2023b) which can be leveraged as potential unknown class information. For example, the foreground of an image showing a bedroom may align with its ground-truth label "bed". However, there are additional objects in the background, such as "curtain" and "lamp", which might not be included in the source domain's label set. This observation motivates us to extract unknown class information inherent in the rich semantics of the source domain data to activate the model's ability to identify unknown class data.

Specifically, we decompose the penultimate-layer image features into fine-grained features and then identify blocks with intrinsic unknown class information. Formally, we denote the feature extractor as $f$ and the classification head as $g$. For source domain data $(x^s, y^s)$, $\mathbf{z}^s = f(x^s) \in \mathbb{R}^{C \times H \times W}$ is the feature map extracted by $f$ without the last global average pooling layer. We further decompose feature map $\mathbf{z}^s$ into $H \times W$ fine-grained granular features $\{\mathbf{Z}_j^s\}_{j=1}^{N=H \times W}$, $\mathbf{Z}_j^s \in \mathbb{R}^C$, representing the features extracted from different regions of the image. Then, we forward these granular features to the classification head $g$ independently to obtain their predictions $\mathrm{argmax}_c\ g_c(\mathbf{Z}_j^s)$, where $g(\cdot) \in \mathbb{R}^{|\mathcal{C}_s|+1}$ denotes the output logit and $g_c(\cdot)$ is the $c$-th element of $g(\cdot)$. Based on whether their predictions align with the ground truth labels $y^s$, we categorize these granular features into known class feature group $\mathcal{Z}_k^s$ and unknown class feature group $\mathcal{Z}_u^s$. In formula,

$$\begin{aligned} \mathcal{Z}_k^s &= \{\mathbf{Z}_j^s |\ \mathrm{argmax}_c\ g_c(\mathbf{Z}_j^s) = y^s\}, \\ \mathcal{Z}_u^s &= \{\mathbf{Z}_j^s |\ \mathrm{argmax}_c\ g_c(\mathbf{Z}_j^s) \neq y^s\}. \end{aligned} \tag{1}$$

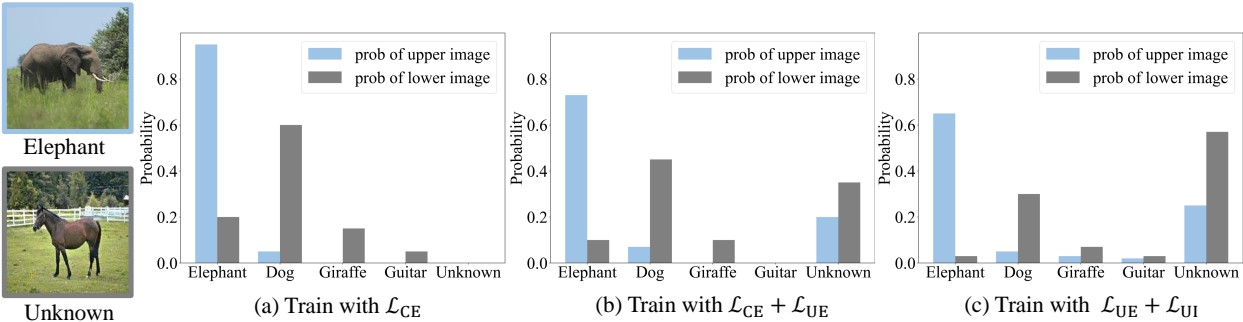

Figure 2: Output probability distributions for input images processed by models trained with different losses. The blue bars represent the probabilities for the upper image from a known class "elephant". The grey bars indicate the probabilities of the lower image, belonging to a category not included in the training data, and thus should be classified as the "unknown" class.

The unknown class features $\mathcal{Z}_u^s$ are then used to activate the classification logit for the unknown class dimension $(|\mathcal{C}_s| + 1)$-th as follows:

$$\mathcal{L}_{\text{UE}}(x_s, y_s) = -\frac{1}{|\mathcal{Z}_u^s|} \sum_{\mathbf{Z}_j^s \in \mathcal{Z}_u^s} \log \frac{\exp(g_{|\mathcal{C}_s|+1}(\mathbf{Z}_j^s))}{\sum_{c \in |\mathcal{C}_s|+1} \exp(g_c(\mathbf{Z}_j^s))}. \tag{2}$$

In other words, we assign the pseudo label $(|\mathcal{C}_s| + 1)$ to these selected granular features, treating them as unknown class data to explicitly activate the logit for the unknown class dimension. As shown in Figure 2(b), when the model is trained with $\mathcal{L}_{\text{CE}} + \mathcal{L}_{\text{UE}}$, the probability for the unknown class is no longer zero. In contrast, when trained with $\mathcal{L}_{\text{CE}}$ (Figure 2(a)), the unknown class probability is always zero for arbitrary inputs.

**Unknown Improvement.** To further increase the probability of the unknown class *without impeding the known class classification performance*, we introduce an unknown improvement loss related to general image classification to enhance the model's response to the unknown class dimension. To differentiate from $\mathbf{z}^s$, we use $\mathbf{z}_{GAP}^s \in \mathbb{R}^C$ to represent the feature extracted by $f$ with the last global average pooling layer. In formula, the unknown improvement loss is:

$$\mathcal{L}_{\text{UI}}(x_s, y_s) = \mathcal{L}_{\text{CE}}(x_s, y_s) - \log \frac{\exp(g_{|\mathcal{C}_s|+1}(\mathbf{z}_{GAP}^s))}{\sum_{c \in |\mathcal{C}_s|+1, c \neq y^s} \exp(g_c(\mathbf{z}_{GAP}^s))} + \lambda \|g(\mathbf{z}_{GAP}^s)\|. \tag{3}$$

In this loss, $\mathcal{L}_{\text{CE}} = -\log \frac{\exp(g_{y^s}(\mathbf{z}_{GAP}^s))}{\sum_{c \in |\mathcal{C}_s|+1} \exp(g_c(\mathbf{z}_{GAP}^s))}$ is the standard cross-entropy loss, which leads the training process and guarantees the known class performance. The second term is designed to enhance the model's classification logit for the unknown class dimension $(|\mathcal{C}_s| + 1)$-th for any input $x^s$, irrespective of its ground-truth label. Note that in the denominator, we exclude the term of the ground truth label $y^s$, thereby preventing degradation of the classification performance for known classes. The last $L_2$ regularization term serves to increase the smoothness of the classifier's output.

**Final Objective for Training.** Hence, our final objective in the training stage is :

$$\mathcal{L}_{\text{UA}}(x_s, y_s) = \mathcal{L}_{\text{UE}}(x_s, y_s) + \mathcal{L}_{\text{UI}}(x_s, y_s). \tag{4}$$

As shown in Figure 2(c), when the model is trained with the full loss $\mathcal{L}_{\text{UE}} + \mathcal{L}_{\text{UI}}$, the real unknown class image is successfully identified (grey bars) since the classification logit of the unknown class dimension is activated by $\mathcal{L}_{\text{UE}}$ and further enhanced by $\mathcal{L}_{\text{UI}}$. Moreover, the known class sample is also classified correctly (blue bars) even though the probability for unknown class dimension is uplifted.

### 3.1.2 Source Condensation

Due to safety and data privacy concerns, SFDA and TTA methods arise to adapt a trained model to the target domain without accessing source training data. However, discarding source data entirely during

adaptation can restrict potential performance gains, as it eliminates the only source of precise supervision. This motivates us to leverage source data while ensuring privacy. We achieve this by condensing the source data into a small dataset that retains class-wise information.

Following recent studies (Wang et al., 2022b; Zhao & Bilen, 2023; Kang et al., 2023), we optimize the condensed dataset $\mathcal{D}_{con} = \cup_{c \in \mathcal{C}_s} \{(\tilde{x}_i, c)\}_{i=1}^{N_{con}}$ by matching the feature distributions of condensed data and source domain data for each known class $c \in \mathcal{C}_s$. More specifically, we utilize the source-trained feature extractor $f$ and freeze it as the backbone for our *Source Condensation*. For each class $c$, we sample a batch of synthesized samples $\tilde{B}_c = \{(\tilde{x}_i, c)\}_{i=1}^{B}$ and real source domain samples $B_c^s = \{(x_i^s, c)\}_{i=1}^{B}$, and minimize the empirical estimate of maximum mean discrepancy between their features extracted by $f$. Moreover, it is known that visual domain has been characterized by image styles, which are commonly described by low-level features' statistics, and instance normalization (IN) normalizes image styles (Ulyanov et al., 2017; Huang & Belongie, 2017; Nam & Kim, 2018). Hence, to obtain a less domain-specific condensed dataset, we apply IN to the low-level features of the source domain data $B_c^s$. We denote $f = f_1 \circ f_2$, where $f_1$ and $f_2$ are the upper and lower parts of $f$ (detailed in Appendix A). The objective is:

$$\mathcal{L}_{\text{SC}}(\mathcal{D}_{con}, \mathcal{D}_s) = \sum_{c=1}^{|\mathcal{C}_s|} \| \frac{1}{|\tilde{B}_c|} \sum_{\tilde{x} \in \tilde{B}_c} f_1 \circ f_2(\tilde{x}) - \frac{1}{|B_c^s|} \sum_{x^s \in B_c^s} f_1 \circ \text{IN} \circ f_2(x^s) \|^2. \tag{5}$$

This process yields a small condensed dataset $\mathcal{D}_{con}$ that retains class-wise information while removing source domain image styles, facilitating the subsequent model adaptation stage.

## 3.2 Open Model Adaptation

Although we tackle the open class problem in the *UASC* stage, deploying the source-trained model in open and new environments remains unsafe and unreliable. We identify two key challenges: (1) *semantic misalignment* arising from the differing distributions of source and target domains; and (2) *asymmetric transfer* caused by the mismatched label sets of source and target domains. To tackle these issues, we propose the *Open Model Adaptation (OMA)* stage, which includes two adjustment terms to facilitate reliable model adaptation in the open-set target domain.

### 3.2.1 Supervisory Guidance with Style Injection

In *Source Condensation*, we obtain a condensed dataset that normalizes image styles. However, the class-wise content information, which is transferable across domains, is preserved. When combined with the style of target domain data, the condensed dataset can serve as synthetic target samples. This motivates us to inject the style information of the target data into the condensed dataset to provide supervisory guidance for known class adaptation. Denote $\tilde{\mathbf{z}}_l = f_2(\tilde{x})$ and $\mathbf{z}_l = f_2(x^t)$ as the low-layer features of condensed data ($\tilde{x}$, $\tilde{y}$) and target data $x^t$ respectively. We conduct style injection (SI) by injecting low-layer feature statistics of the target data into the condensed data:

$$\text{SI}(\tilde{\mathbf{z}}_l) = \mu(\mathbf{z}_l) + \sigma(\mathbf{z}_l) \cdot \frac{\tilde{\mathbf{z}}_l - \mu(\tilde{\mathbf{z}}_l)}{\sigma(\tilde{\mathbf{z}}_l)}, \tag{6}$$

where $\mu(\cdot)$ and $\sigma(\cdot)$ are the channel-wise mean and standard deviation of the low-layer features, respectively. In formula, they are calculated in terms of a feature map $\mathbf{z} \in \mathbb{R}^{C' \times H' \times W'}$ as follows:

$$\mu(\mathbf{z}) = \frac{1}{H'W'} \sum_{h=1}^{H'} \sum_{w=1}^{W'} \mathbf{z}_{:,h,w}$$

$$\sigma(\mathbf{z}) = \sqrt{\frac{1}{H'W'} \sum_{h=1}^{H'} \sum_{w=1}^{W'} (\mathbf{z}_{:,h,w} - \mu(\mathbf{z}))^2} \tag{7}$$

Then, we utilize the style-injected feature $\tilde{\mathbf{z}}_{\mathrm{SI}} = f_1(\mathrm{SI}(\tilde{\mathbf{z}}_l))$ of the condensed data to provide supervisory guidance for model adaptation:

$$\mathcal{L}_{\mathrm{SI}}(\tilde{x}, \tilde{y}) = -\log \frac{\exp(g_{\tilde{y}}(\tilde{\mathbf{z}}_{\mathrm{SI}}))}{\sum_{c \in |\mathcal{C}_s|+1} \exp(g_c(\tilde{\mathbf{z}}_{\mathrm{SI}}))} \tag{8}$$

where $\tilde{y}$ is the supervisory label of condensed data $\tilde{x}$. Intuitively, style-injected condensed data exhibit target domain style and known class content, making them suitable for providing supervisory guidance for known class adaptation. In other words, we treat the target-style injected condensed data as virtual known class data from the target domain, and adapt the model to recognize them during the test phase.

### 3.2.2 Decoupled Domain Alignment

Asymmetric label sets of source and target domains cause mismatches for known and unknown classes. We propose to align the cross-domain knowledge for both known and unknown classes. The core idea is to adapt the model to be insensitive to domain variations when recognizing known and unknown classes.

In the *OMA* stage, we have access to the condensed dataset $\mathcal{D}_{con}$ and unlabeled target domain data $\mathcal{D}_t$. We decompose the feature maps (extracted by feature extractor $f$ without the global average pooling layer) of condensed data $\tilde{\mathbf{z}} = f(\tilde{x}) \in \mathbb{R}^{C \times H \times W}$ and target data $\mathbf{z}^t = f(x^t) \in \mathbb{R}^{C \times H \times W}$ into granular features $\{\tilde{\mathbf{Z}}_i\}_{i=1}^{N=H \times W} \in \mathbb{R}^C$ and $\{\mathbf{Z}_j^t\}_{j=1}^{N=H \times W} \in \mathbb{R}^C$, respectively. For each granular feature $\mathbf{Z}$, we forward it to the classifier $g$ to obtain its probabilities for being known and unknown classes as follows:

$$\begin{aligned} w_k(\mathbf{Z}) &= \frac{\sum_{c \in |\mathcal{C}_s|} \exp(g_c(\mathbf{Z}))}{\sum_{c' \in |\mathcal{C}_s|+1} \exp(g_{c'}(\mathbf{Z}))}, \\ w_u(\mathbf{Z}) &= \frac{\exp(g_{|\mathcal{C}_s|+1}(\mathbf{Z}))}{\sum_{c' \in |\mathcal{C}_s|+1} \exp(g_{c'}(\mathbf{Z}))}. \end{aligned} \tag{9}$$

Then, we introduce two lightweight binary domain classifiers $g_k$ and $g_u$, which are designed to discern whether the granular features (classified as known and unknown classes) are from the condensed data or target data respectively. We achieve cross-domain knowledge alignment by raising the model's insensitivity of domain discrepancies. More specifically, we denote $\mathrm{D} = \{0, 1\}$ as the domain label, referring that granular feature $\mathbf{Z}$ is from the condensed data (label 0) or target data (label 1). Let $\mathrm{p}_{0,k}(\mathbf{Z})$ denote the probability that the known-class domain classifier $g_k$ classifies the granular feature $\mathbf{Z}$ as belonging to the condensed dataset (domain label 0), and $\mathrm{p}_{1,k}(\mathbf{Z})$ denote the probability that $\mathbf{Z}$ belongs to the target domain (domain label 1). Similarly, $\mathrm{p}_{0,u}(\mathbf{Z})$ and $\mathrm{p}_{1,u}(\mathbf{Z})$ are the corresponding probabilities obtained after forwarding the unknown-class domain classifier $g_u$. The domain alignment loss is a weighted binary cross-entropy loss:

$$\begin{aligned} \mathcal{L}_{\mathrm{DA}}(\tilde{x}, x_t) = -\frac{1}{HW} \sum_{\mathbf{Z} \in \{\tilde{\mathbf{Z}}_i\} \cup \{\mathbf{Z}_j^t\}} &\{w_k(\mathbf{Z})[\mathrm{D}\log(p_{o,k}(\mathbf{Z})) + (1-\mathrm{D})\log(p_{1,k}(\mathbf{Z}))] + \\ &w_u(\mathbf{Z})[\mathrm{D}\log(p_{o,u}(\mathbf{Z})) + (1-\mathrm{D})\log(p_{1,u}(\mathbf{Z}))]\}. \end{aligned} \tag{10}$$

For granular features from the condensed data $\mathbf{Z} \in \{\tilde{\mathbf{Z}}_i\}_{i=1}^{N=H \times W}$ (domain label $D = 0$), $\mathcal{L}_{\mathrm{DA}}$ raises the probabilities $((p_{1,k}(\mathbf{Z})$ and $p_{1,u}(\mathbf{Z}))$ for being classified as from the target data. Similarly, for granular features from the target data $\mathbf{Z} \in \{\mathbf{Z}_j^t\}_{j=1}^{N=H \times W}$ (domain label $D = 1$), $\mathcal{L}_{\mathrm{DA}}$ increases their probabilities $((p_{0,k}(\mathbf{Z})$ and $p_{0,u}(\mathbf{Z}))$ for being classified as from the condensed data. In this way, we raise the model's insensitivity to the domain variances, enforcing the model to align cross-domain knowledge for both known and unknown classes.

**Final Objective for Adaptation.** Hence, the final adaptation loss for our *OMA* stage is:

$$\mathcal{L}_{\mathrm{MA}}(\tilde{x}, \tilde{y}, x_t) = \mathcal{L}_{\mathrm{DA}}(\tilde{x}, x_t) + \alpha \mathcal{L}_{\mathrm{SI}}(\tilde{x}, \tilde{y}), \tag{11}$$

where $\alpha$ is a balancing hyper-parameter. Intuitively, $\mathcal{L}_{\mathrm{SI}}$ enhances the performance for known class classification performance, while $\mathcal{L}_{\mathrm{DA}}$ facilitates the model to identify unknown class data in the open-set target domain. These two terms work collaboratively to update the feature extractor $f$, main classifier $g$, along with two lightweight binary domain classifiers $g_k$ and $g_u$ during the *OMA* stage.

## 4 Experiments

### 4.1 Experimental Setup

**Benchmarks.** We conduct extensive experiments on three standard DG benchmarks to validate the effectiveness of our proposed ADA: (1) OFFICE-HOME (Venkateswara et al., 2017) contains images from 65 classes across four domains (Artistic, Clipart, Product, and Real World). The first 15 alphabetically ordered classes are designated as known classes $\mathcal{C}_s$ and the subsequent 50 classes are treated as the unknown class $\mathcal{C}_t^u$. (2) OFFICE-31 (Saenko et al., 2010) comprises 31 classes of images sourced from three domains (Amazon, DSLR, and Webcam). The ten classes shared by OFFICE-31 and CALTECH-256 (Gong et al., 2012) are adopted as $\mathcal{C}_s$ and the alphabetically last 11 classes are $\mathcal{C}_t^u$. (3) PACS (Li et al., 2017) consists of images from four domains (Photo, Art Painting, Cartoon, and Sketch) with distinct styles. It has seven classes in total. Four classes (dog, elephant, giraffe, and guitar) constitute $\mathcal{C}_s$, and other classes are $\mathcal{C}_t^u$.

**Evaluation Protocols.** Following prior works (Bucci et al., 2020; Chen et al., 2023a), we adopt the h-score ($hs$) as the key evaluation metric, which emphasizes importance for both known class accuracy $acc_k$ and unknown class accuracy $acc_u$. More specifically, $hs$ is calculated as $2 \times \frac{acc_k \times acc_u}{acc_k + acc_u}$, and a high $hs$ requires both $acc_k$ and $acc_u$ to be high and balanced.

**Implementation Details.** For all experiments, we use the ImageNet pre-trained ResNet-50 (He et al., 2016) as the feature extractor and two fully connected layers as the classification head, which is consistent with previous works (Yang et al., 2022b; Liang et al., 2020) for a fair comparison. The binary domain classification head in the *OMA* stage consists of two Linear-BN-LeakyReLU blocks and a linear layer. During training, we use SGD with a momentum of 0.9 as the optimizer, and the batch size is set as 64. The learning rate is set as 1e-3 for the backbone and 1e-2 for the classifier, and the training epoch is set to 30 for all datasets. In the *OMA* stage, the fine-tune learning rate is reduced to one-tenth of the learning rate adopted in the training stage. More details are provided in Appendix A.

### 4.2 Baselines

We conduct experiments in both online and offline settings. For the online setting, we compare ADA with TTA methods, including Tent (Wang et al., 2020), OSTTA (Lee et al., 2023), UniEnt (Gao et al., 2024), and ART (Chen et al., 2023b). Note that except for Tent, others are specifically developed for the open-set setting. For the offline setting, we compare with SFDA and OSDG baselines, including SHOT (Liang et al., 2020), AaD (Yang et al., 2022b), and OneRing (Yang et al., 2022c). Some methods such as Tent are incapable of directly handling the open-set scenario, and we adapt these methods for the open-set scenario by calculating their predicted probabilities' entropy and predefining an entropy threshold (Zhu & Li, 2022). Specifically, we set the threshold as $0.5 * \log |\mathcal{C}_s|$, and the data is classified as unknown class when the entropy of the probability distribution is larger than predefined threshold. Details are provided in Appendix A.

### 4.3 Main Results

We compare our method with baselines in both offline and online settings. The detailed results are reported in Table 2 (OFFICE-HOME and PACS) and Table 3 (OFFICE-31). We elaborate our findings as follows.

**Offline Setting.** In the offline setting, the model makes predictions after being fine-tuned on the entire unlabeled target dataset. The results indicate that ADA exhibits a significant performance advantage over baseline methods across all three benchmarks. For instance, ADA outperforms the best baseline method in $hs$ by 4.9% for OFFICE-HOME, 7.3% for PACS, and 4.6% for OFFICE-31. More specifically, we point out two key findings: (1) ADA consistently delivers high $hs$ across individual tasks by effectively balancing both $acc_k$ and $acc_u$. In contrast, other methods such as SHOT often yield imbalanced results on some adaptation tasks, thus leading to inferior $hs$. This is likely due to their reliance on the additional step of predefining a threshold for distinguishing the unknown class. The optimal threshold can vary significantly across individual tasks, making it difficult to set a consistently optimal value. ADA introduces the additional unknown class dimension $(|\mathcal{C}_s| + 1)$-th to the classifier, thereby avoiding this issue. (2) The performance of baseline methods varies across different benchmarks. For instance, SHOT only brings marginal improvement compared to ERM on

Table 2: Results of all experiments for OFFICE-HOME and PACS. We highlight the **best** and second results.

**OFFICE-HOME**

| Setting | Methods | A2C acc_k | A2C acc_u | A2C hs | A2P acc_k | A2P acc_u | A2P hs | A2R acc_k | A2R acc_u | A2R hs | C2A acc_k | C2A acc_u | C2A hs | C2P acc_k | C2P acc_u | C2P hs | C2R acc_k | C2R acc_u | C2R hs |
|---|---|---|---|---|---|---|---|---|---|---|---|---|---|---|---|---|---|---|---|
| Online | Tent | 61.7 | 50.3 | 55.4 | 77.5 | 48.1 | 59.4 | 88.5 | 45.3 | 59.9 | 67.9 | 52.8 | 59.4 | 75.7 | 42.2 | 54.2 | 82.5 | 43.4 | 56.8 |
| | OSTTA | 52.7 | 65.0 | 58.2 | 63.5 | 63.4 | 63.5 | 72.3 | 55.5 | 62.8 | 55.2 | 67.4 | 60.7 | 63.2 | 69.9 | 66.4 | 72.2 | 71.9 | 72.0 |
| | UniEnt | 63.1 | 48.6 | 54.9 | 78.6 | 52.5 | 62.9 | 89.0 | 45.2 | 59.9 | 68.4 | 51.1 | 58.5 | 76.5 | 40.8 | 53.2 | 78.9 | 45.3 | 57.6 |
| | ART | 57.0 | 54.8 | 55.9 | 78.8 | 52.1 | 62.8 | 87.5 | 61.6 | 72.3 | 63.4 | 65.6 | 64.5 | 76.4 | 50.4 | 60.8 | 81.1 | 63.0 | 70.9 |
| | **ADA** | 52.2 | 80.7 | 63.4 | 71.4 | 74.3 | 72.8 | 84.4 | 73.9 | 78.8 | 42.9 | 87.2 | 57.6 | 58.4 | 76.7 | 66.3 | 65.4 | 79.6 | 71.8 |
| Offline | ERM | 35.5 | 76.1 | 48.4 | 48.5 | 62.3 | 54.5 | 61.7 | 51.2 | 56.0 | 42.1 | 64.7 | 51.0 | 52.0 | 47.1 | 49.4 | 61.8 | 52.1 | 56.5 |
| | SHOT | 71.7 | 31.6 | 43.9 | 88.9 | 26.6 | 41.0 | 91.8 | 34.6 | 50.3 | 74.2 | 50.7 | 60.3 | 82.2 | 27.2 | 40.8 | 86.5 | 34.8 | 49.6 |
| | AaD | 51.2 | 61.3 | 55.8 | 66.8 | 56.7 | 61.3 | 82.6 | 63.8 | 72.0 | 60.0 | 77.1 | 67.5 | 67.8 | 60.7 | 64.0 | 76.4 | 65.9 | 70.8 |
| | OneRing | 50.7 | 76.4 | 61.0 | 75.3 | 64.8 | 69.7 | 86.9 | 64.8 | 74.2 | 55.0 | 77.0 | 64.2 | 64.8 | 67.7 | 66.2 | 72.3 | 60.8 | 66.1 |
| | **ADA** | 55.8 | 80.1 | 65.8 | 72.1 | 75.3 | 73.3 | 84.0 | 74.8 | 79.1 | 45.9 | 83.1 | 59.1 | 62.0 | 73.2 | 67.1 | 70.6 | 75.5 | 73.0 |

| Methods | P2A acc_k | P2A acc_u | P2A hs | P2C acc_k | P2C acc_u | P2C hs | P2R acc_k | P2R acc_u | P2R hs | R2A acc_k | R2A acc_u | R2A hs | R2C acc_k | R2C acc_u | R2C hs | R2P acc_k | R2P acc_u | R2P hs | Average acc_k | Average acc_u | Average hs |
|---|---|---|---|---|---|---|---|---|---|---|---|---|---|---|---|---|---|---|---|---|---|
| Tent | 62.9 | 60.6 | 61.7 | 58.8 | 53.6 | 56.1 | 85.7 | 47.6 | 61.2 | 75.1 | 52.7 | 62.0 | 63.1 | 51.9 | 56.9 | 85.8 | 46.6 | 60.4 | 73.8 | 49.6 | 58.6 |
| OSTTA | 47.2 | 81.8 | 59.9 | 47.4 | 75.6 | 58.3 | 75.7 | 61.2 | 67.7 | 64.1 | 72.9 | 68.2 | 53.9 | 74.6 | 62.5 | 80.1 | 70.8 | 75.2 | 62.3 | 69.2 | 64.6 |
| UniEnt | 63.7 | 58.7 | 61.1 | 60.0 | 50.0 | 54.6 | 84.6 | 56.4 | 67.6 | 77.4 | 49.4 | 60.3 | 64.3 | 49.1 | 55.6 | 84.0 | 57.7 | 68.4 | **74.1** | 50.4 | 59.6 |
| ART | 63.5 | 48.4 | 54.9 | 55.5 | 54.3 | 54.9 | 87.5 | 52.2 | 65.4 | 59.2 | 52.8 | 55.8 | 87.7 | 54.2 | 67.0 | 74.6 | 57.0 | 64.6 | 63.2 | 57.2 | 62.5 |
| **ADA** | 42.9 | 83.5 | 56.7 | 43.1 | 79.9 | 56.0 | 76.5 | 77.8 | 77.1 | 59.5 | 79.0 | 67.9 | 49.0 | 78.5 | 60.3 | 79.5 | 75.2 | 77.3 | 60.4 | **78.9** | **67.2** |
| ERM | 39.3 | 68.6 | 50.0 | 32.2 | 44.3 | 37.3 | 62.1 | 65.0 | 63.5 | 45.1 | 44.9 | 45.0 | 44.6 | 36.2 | 40.0 | 76.2 | 37.0 | 49.8 | 50.0 | 54.1 | 50.1 |
| SHOT | 72.1 | 54.9 | 62.4 | 64.7 | 30.8 | 41.7 | 89.6 | 39.0 | 54.3 | 77.1 | 52.4 | 62.4 | 64.5 | 37.1 | 47.1 | 86.9 | 34.6 | 49.4 | **79.2** | 37.9 | 50.3 |
| AaD | 54.9 | 80.5 | 65.3 | 50.4 | 63.9 | 56.4 | 77.6 | 68.4 | 72.7 | 61.8 | 74.3 | 67.5 | 54.9 | 59.7 | 57.2 | 76.6 | 59.1 | 66.7 | 65.1 | 65.9 | 64.8 |
| OneRing | 54.0 | 75.2 | 62.8 | 43.9 | 61.2 | 51.1 | 69.5 | 68.9 | 69.2 | 70.8 | 65.8 | 68.2 | 49.0 | 70.8 | 57.9 | 85.0 | 61.7 | 71.5 | 64.8 | 67.9 | 65.2 |
| **ADA** | 44.6 | 79.5 | 57.1 | 49.5 | 75.4 | 59.7 | 76.5 | 77.7 | 77.1 | 63.1 | 77.3 | 69.5 | 53.1 | 73.7 | 61.7 | 79.5 | 75.2 | 77.3 | 63.1 | **76.7** | **68.4** |

**PACS**

| Setting | Methods | A2C acc_k | A2C acc_u | A2C hs | A2P acc_k | A2P acc_u | A2P hs | A2S acc_k | A2S acc_u | A2S hs | C2A acc_k | C2A acc_u | C2A hs | C2P acc_k | C2P acc_u | C2P hs | C2S acc_k | C2S acc_u | C2S hs |
|---|---|---|---|---|---|---|---|---|---|---|---|---|---|---|---|---|---|---|---|
| Online | Tent | 53.1 | 43.6 | 47.9 | 40.0 | 43.6 | 41.7 | 64.7 | 40.6 | 49.9 | 74.3 | 30.1 | 42.8 | 75.7 | 28.6 | 41.6 | 69.5 | 32.8 | 44.5 |
| | OSTTA | 54.2 | 55.5 | 54.8 | 44.2 | 43.6 | 43.9 | 43.1 | 52.4 | 47.3 | 62.2 | 46.1 | 52.9 | 72.6 | 39.0 | 50.7 | 60.0 | 55.5 | 57.7 |
| | UniEnt | 55.5 | 41.5 | 47.5 | 42.2 | 47.2 | 44.6 | 67.1 | 43.0 | 52.4 | 64.8 | 30.5 | 41.5 | 89.6 | 28.1 | 42.8 | 70.1 | 32.9 | 44.7 |
| | ART | 52.3 | 70.2 | 59.9 | 27.4 | 39.3 | 32.3 | 50.7 | 51.4 | 51.1 | 57.3 | 63.8 | 60.4 | 87.0 | 67.6 | 76.1 | 60.6 | 73.9 | 66.6 |
| | **ADA** | 64.1 | 60.0 | 62.0 | 29.9 | 53.0 | 38.3 | 28.9 | 73.3 | 41.5 | 95.7 | 69.5 | 80.5 | 59.2 | 50.6 | 54.6 | 55.0 | 51.6 | 53.3 |
| Offline | ERM | 38.4 | 80.0 | 51.9 | 95.3 | 90.0 | 92.6 | 42.3 | 54.0 | 47.5 | 36.6 | 75.9 | 49.4 | 64.5 | 86.3 | 73.8 | 33.0 | 92.9 | 48.7 |
| | SHOT | 82.7 | 32.7 | 46.9 | 99.0 | 32.6 | 49.1 | 76.4 | 32.8 | 45.9 | 80.8 | 34.2 | 48.1 | 96.7 | 45.8 | 62.1 | 61.6 | 31.3 | 41.5 |
| | AaD | 74.9 | 44.4 | 55.8 | 78.3 | 52.8 | 63.1 | 54.5 | 30.5 | 39.1 | 40.8 | 42.2 | 41.5 | 56.0 | 35.2 | 43.3 | 31.4 | 26.4 | 28.7 |
| | OneRing | 65.4 | 38.0 | 48.1 | 98.7 | 55.4 | 71.0 | 53.4 | 38.7 | 44.9 | 54.2 | 47.6 | 50.7 | 73.2 | 41.8 | 53.2 | 60.3 | 55.6 | 57.8 |
| | **ADA** | 58.8 | 55.3 | 57.0 | 40.6 | 75.3 | 52.8 | 19.2 | 82.7 | 31.2 | 95.4 | 88.0 | 91.6 | 64.0 | 61.9 | 62.9 | 62.4 | 57.6 | 59.9 |

| Methods | P2A acc_k | P2A acc_u | P2A hs | P2C acc_k | P2C acc_u | P2C hs | P2S acc_k | P2S acc_u | P2S hs | S2A acc_k | S2A acc_u | S2A hs | S2C acc_k | S2C acc_u | S2C hs | S2P acc_k | S2P acc_u | S2P hs | Average acc_k | Average acc_u | Average hs |
|---|---|---|---|---|---|---|---|---|---|---|---|---|---|---|---|---|---|---|---|---|---|
| Tent | 77.1 | 58.1 | 66.3 | 50.3 | 66.3 | 57.2 | 42.0 | 64.5 | 50.9 | 67.8 | 49.5 | 57.2 | 69.9 | 38.6 | 49.8 | 55.0 | 55.0 | 55.0 | 61.6 | 45.9 | 50.4 |
| OSTTA | 68.3 | 69.5 | 68.9 | 47.0 | 34.4 | 39.7 | 33.7 | 72.8 | 46.1 | 68.3 | 71.5 | 69.9 | 53.1 | 65.3 | 58.6 | 65.0 | 33.9 | 44.5 | 56.0 | 53.3 | 52.9 |
| UniEnt | 77.7 | 57.8 | 66.3 | 57.1 | 64.5 | 60.6 | 44.6 | 61.2 | 51.6 | 78.3 | 49.4 | 60.6 | 72.4 | 38.8 | 50.6 | 91.4 | 55.4 | 69.0 | **67.6** | 45.9 | 52.7 |
| ART | 70.0 | 79.2 | 74.3 | 23.5 | 61.1 | 33.9 | 29.9 | 57.5 | 39.3 | 29.8 | 46.3 | 36.3 | 18.6 | 42.1 | 25.8 | 89.0 | 75.7 | 81.8 | 49.7 | 60.7 | 53.2 |
| **ADA** | 71.3 | 72.7 | 72.0 | 51.1 | 54.7 | 52.9 | 51.2 | 68.7 | 58.7 | 25.7 | 86.9 | 39.7 | 17.4 | 89.4 | 29.1 | 59.6 | 62.1 | 60.8 | 50.8 | **66.0** | **53.6** |
| ERM | 43.2 | 95.2 | 59.5 | 12.5 | 82.4 | 21.7 | 14.7 | 99.5 | 25.6 | 16.7 | 44.3 | 24.2 | 21.2 | 41.0 | 28.0 | 24.8 | 29.6 | 27.0 | 36.9 | **72.6** | 45.8 |
| SHOT | 80.1 | 37.5 | 51.1 | 74.7 | 31.6 | 44.4 | 73.5 | 20.3 | 31.8 | 47.8 | 32.9 | 39.0 | 71.9 | 39.3 | 50.8 | 95.2 | 41.0 | 57.3 | **78.4** | 34.3 | 47.3 |
| AaD | 60.0 | 48.1 | 53.4 | 63.0 | 47.7 | 54.3 | 66.6 | 54.9 | 60.2 | 73.1 | 70.6 | 71.8 | 71.0 | 58.9 | 64.4 | 85.5 | 70.2 | 77.1 | 62.9 | 48.5 | 54.4 |
| OneRing | 71.5 | 64.1 | 67.6 | 32.1 | 57.9 | 41.3 | 30.5 | 59.1 | 40.2 | 39.1 | 86.3 | 53.8 | 54.0 | 62.9 | 58.1 | 56.0 | 90.3 | 69.1 | 57.4 | 58.2 | 54.7 |
| **ADA** | 68.4 | 82.4 | 74.8 | 49.7 | 67.6 | 57.3 | 48.9 | 75.8 | 59.4 | 37.3 | 76.7 | 50.2 | 32.5 | 83.0 | 46.7 | 69.6 | 54.2 | 60.9 | 53.9 | 71.7 | **58.7** |

OFFICE-HOME and PACS. In contrast, ADA consistently delivers enhanced results on all three benchmarks, highlighting the significance of our designed two stages.

**Online Setting.** In online setting, target data continuously arrives, and the model adapts on it and makes prediction presently. We compare ADA with TTA baselines, and it shows that ADA surpasses baselines on three

Table 3: Results of all experiments for OFFICE-31. We highlight the **best** and second results.

| | | OFFICE-31 | | | | | | | | | | | | | | | | | | | |
|---|---|---|---|---|---|---|---|---|---|---|---|---|---|---|---|---|---|---|---|---|---|
| **Methods** | A2D | | | A2W | | | D2A | | | D2W | | | W2A | | | W2D | | | **Average** | | |
| | $acc_k$ | $acc_u$ | $hs$ | $acc_k$ | $acc_u$ | $hs$ | $acc_k$ | $acc_u$ | $hs$ | $acc_k$ | $acc_u$ | $hs$ | $acc_k$ | $acc_u$ | $hs$ | $acc_k$ | $acc_u$ | $hs$ | $acc_k$ | $acc_u$ | $hs$ |
| Tent | 82.6 | 54.6 | 65.8 | 86.9 | 57.7 | 69.3 | 61.4 | 89.0 | 72.6 | 88.3 | 81.7 | 84.9 | 79.4 | 75.7 | 77.5 | 95.0 | 77.7 | 85.5 | 82.2 | 72.7 | 75.9 |
| OSTTA | 81.2 | 83.0 | 82.1 | 75.1 | 80.2 | 77.5 | 34.8 | 98.4 | 51.4 | 83.8 | 96.2 | 89.6 | 63.2 | 92.5 | 75.1 | 97.2 | 93.6 | 95.4 | 72.6 | **90.6** | 78.5 |
| UniEnt | 90.3 | 60.6 | 72.5 | 86.9 | 57.3 | 69.1 | 64.3 | 87.0 | 73.9 | 98.3 | 81.7 | 89.2 | 80.6 | 74.9 | 77.7 | 98.2 | 77.1 | 86.4 | **86.4** | 73.1 | 78.1 |
| ART | 91.0 | 66.5 | 76.8 | 86.2 | 63.7 | 73.2 | 56.3 | 90.7 | 69.4 | 68.0 | 86.1 | 76.0 | 76.1 | 81.9 | 78.9 | 95.8 | 79.3 | 86.8 | 78.9 | 78.0 | 76.9 |
| **ADA** (Online) | 88.2 | 79.5 | 83.6 | 76.3 | 73.2 | 74.7 | 62.7 | 88.6 | 73.4 | 94.1 | 82.7 | 88.0 | 72.8 | 85.4 | 78.6 | 99.0 | 85.9 | 92.0 | 82.2 | 82.5 | **81.7** |
| ERM | 79.9 | 71.9 | 75.7 | 78.1 | 82.0 | 80.0 | 13.0 | 90.2 | 22.7 | 18.2 | 95.0 | 30.5 | 30.8 | 98.1 | 46.9 | 88.2 | 93.6 | 90.8 | 51.3 | **88.5** | 57.8 |
| SHOT | 90.5 | 38.7 | 54.2 | 91.4 | 51.2 | 65.6 | 83.7 | 33.3 | 47.7 | 96.3 | 78.6 | 86.6 | 77.0 | 44.6 | 56.5 | 93.0 | 82.8 | 87.6 | **88.7** | 54.8 | 66.3 |
| AaD | 75.0 | 75.3 | 75.2 | 78.5 | 74.6 | 76.5 | 72.6 | 67.9 | 70.2 | 90.2 | 86.4 | 88.3 | 70.8 | 72.4 | 71.6 | 94.3 | 91.3 | 92.8 | 80.2 | 78.0 | 79.1 |
| OneRing | 81.5 | 70.9 | 75.8 | 76.5 | 56.8 | 65.2 | 69.4 | 88.8 | 77.9 | 85.0 | 70.3 | 76.9 | 73.7 | 85.8 | 79.3 | 89.2 | 86.6 | 87.9 | 79.2 | 76.5 | 77.2 |
| **ADA** (Offline) | 88.2 | 80.3 | 84.0 | 75.4 | 73.6 | 74.5 | 75.5 | 82.0 | 78.7 | 95.6 | 80.8 | 87.6 | 74.5 | 83.4 | 78.7 | 97.9 | 87.8 | 92.6 | 84.5 | 81.3 | **82.7** |

benchmarks consistently. Compared to previous best methods, `ADA` improves the $hs$ by 4.0% for OFFICE-HOME, 0.8% for PACS, and 4.1% for OFFICE-31. Furthermore, we observe a slight decrease in `ADA` in the online setting compared to the performance in the offline setting. This can be attributed to the "one-pass" nature of the online adaptation setting. Nevertheless, compared to existing open-set TTA methods such as OSTTA, UniEnt, and ART, `ADA` demonstrates improvements across all benchmarks. These results suggest that `ADA` is effective for both online and offline settings.

### 4.4 Ablation and More Studies

**Ablation on Two Stages.** Table 4 shows the ablation results of the proposed two stages *UASC* and *OMA*. It can be observed that the combination of the two stages outperforms all other variants. Moreover, we find that even with the *UASC* training stage alone, our method achieves competitive results. This suggests that *UASC* is effective in enabling the model to identify unknown class data. To further examine the efficacy of the *OMA* stage, we integrate *UASC* with existing SFDA and TTA methods that address the domain shift problem, including SHOT (Liang et al., 2020), Tent (Wang et al., 2020), TTT (Sun et al., 2020), and UniEnt (Gao et al., 2024). As shown in the table, *UASC* exhibits worse performance when combined with SHOT, TTT, and Tent, and a marginal improvement is observed when combined with UniEnt. These findings substantiate that collaboration of the *UASC* and *OMA* stages maximizes the effectiveness of `ADA`.

Table 4: Ablation results ($hs$, %) on three benchmarks.

| **Methods** | OFFICE-HOME | OFFICE-31 | PACS | **Average** |
|---|---|---|---|---|
| $\mathcal{L}_{\text{UE}}$ | 65.1 | 72.3 | 48.6 | 62.0 |
| $\mathcal{L}_{\text{UI}}$ | 63.8 | 64.5 | 48.0 | 58.8 |
| $\mathcal{L}_{\text{UE}} + \mathcal{L}_{\text{UI}}$ (*UASC*) | 66.0 | 80.1 | 51.8 | 66.0 |
| $UASC+\mathcal{L}_{\text{DA}}$ | 65.5 | 80.3 | 53.6 | 66.5 |
| $UASC+\mathcal{L}_{\text{SI}}$ | 66.1 | 81.8 | 53.1 | 67.0 |
| $UASC+$SHOT | 61.4 | 74.5 | 46.4 | 60.8 |
| $UASC+$Tent | 63.1 | 77.8 | 53.6 | 64.8 |
| $UASC+$TTT | 62.8 | 80.4 | 47.9 | 63.7 |
| $UASC+$UniEnt | 66.2 | 80.3 | 53.4 | 66.6 |
| $UASC+\mathcal{L}_{\text{DA}}+\mathcal{L}_{\text{SI}}$ (*UASC+OMA*) | 68.4 | 82.7 | 58.7 | **69.9** |

**Analysis on Unknown Activation.** In the *UASC* stage, we design $\mathcal{L}_{\text{UE}}$ and $\mathcal{L}_{\text{UI}}$ to activate and enhance the model's response for the unknown class. The results of their individual effects are shown in Table 4. It shows that when these two losses are combined, the model exhibits better performance than their individuals. We present an illustration in Figure 2, showing the probability distributions when inputs pass models trained with different losses. It shows that when the model is trained with the full loss $\mathcal{L}_{\text{UE}} + \mathcal{L}_{\text{UI}}$, it effectively distinguishes unknown class data while maintaining its performance for classifying known class data.

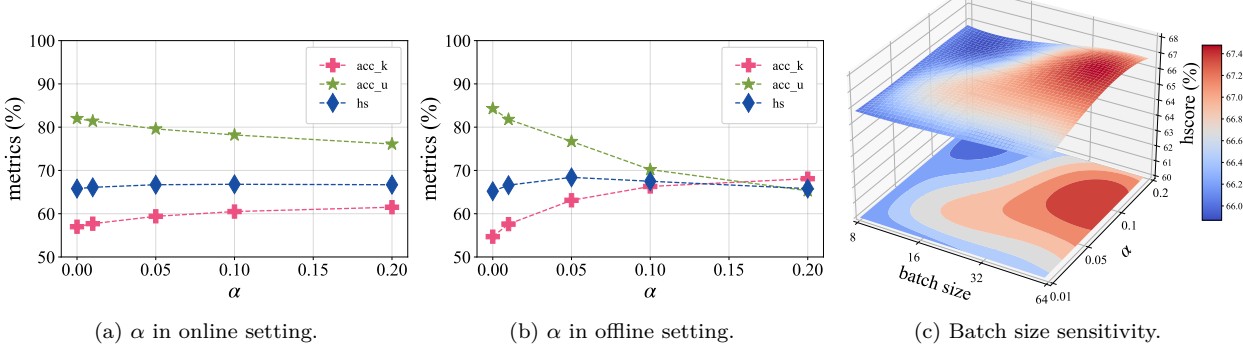

(a) $\alpha$ in online setting.  (b) $\alpha$ in offline setting.  (c) Batch size sensitivity.

Figure 3: (a)-(b) Sensitivity of $\alpha$ for online and offline settings respectively. (c) Sensitivity of test batch size for online setting. All results are on OFFICE-HOME.

**Analysis on Open Model Adaptation.** The ablation results of two losses $\mathcal{L}_{\mathrm{DA}}$ and $\mathcal{L}_{\mathrm{SI}}$ proposed in *OMA* are provided in Table 4, and it shows that each loss contributes marginally, while the joint of them achieves better results. Moreover, we study the sensitivity of the hyper-parameter $\alpha$ in Equation 11. In Figure 3(a)(b), we can observe that as $\alpha$ increases, $acc_k$ increases while $acc_u$ decreases in both online and offline settings. This aligns with our intuition that $\mathcal{L}_{\mathrm{SI}}$ enhances known class performance and $\mathcal{L}_{\mathrm{DA}}$ strengthens unknown class identification. When $\alpha$ is around 0.05 to 0.1, a balance between $acc_k$ and $acc_u$ is achieved, resulting in the highest $hs$. Furthermore, our analysis reveals that the results are more stable to variations in $\alpha$ in the online setting compared to the offline setting. This can be explained by the fact that in online adaptation, the model updates and predicts incrementally as it continuously receives data from target domain. Hence, the model's adjustments are more gradual and controlled in the online setting than in the offline setting.

We also investigate the influence of varying test batch sizes in the online adaptation context. The result is shown in Figure 3(c). It can be observed that $hs$ is not significantly affected by the variation of batch size and $\alpha$. This insensitivity underscores the consistent performance of `ADA` across different conditions, highlighting its robustness in a variety of scenarios.

**Analysis on Source Condensation.** We empirically study the impact of the condensed dataset. Our analysis focuses on two aspects. First, we assess the benefits of employing a condensed dataset over the raw training data. We explore two alternative strategies: randomly sampling a subset from the source training data, and selecting samples that are closest to the class prototype for each class. As shown in Table 5, the strategy of employing a condensed dataset outperforms the other two variants, indicating that the condensed dataset offers more comprehensive information than a selected subset of the source data. Notably, the performance of all three strategies exceeds that of the scenario where the *OMA* stage is ablated (51.8% as shown in Table 4), underscoring the importance of the *OMA* stage. Moreover, using the original source data carries a risk of data privacy leakage, whereas source condensation avoids this issue.

Table 5: Impacts of condensed dataset and different sampling strategies. $hs$ for PACS are reported.

| Strategies | Random Selection | Class Prototype | Condensed Dataset |
|:---:|:---:|:---:|:---:|
| $hs$ | 56.9 | 56.7 | 58.7 |

Second, we investigate the impact of the number of condensed data. As shown in Figure 4(a), `ADA` remains effective when the number of condensed data per class is at least 10. This indicates a small condensed dataset can preserve enough source domain information, reducing the computation burden of the *OMA* stage.

**Feature Visualization.** We present the GradCAM (Selvaraju et al., 2017) visualization of `ADA` and ERM in Figure 4(b). Note that although ERM does not have supervision for the unknown class training, we heuristically visualize its $(|\mathcal{C}_s|+1)$-th output to explore its response to unknown-class inputs, which serves as

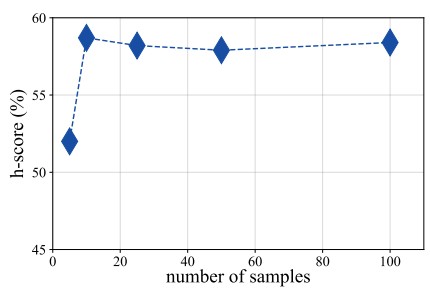 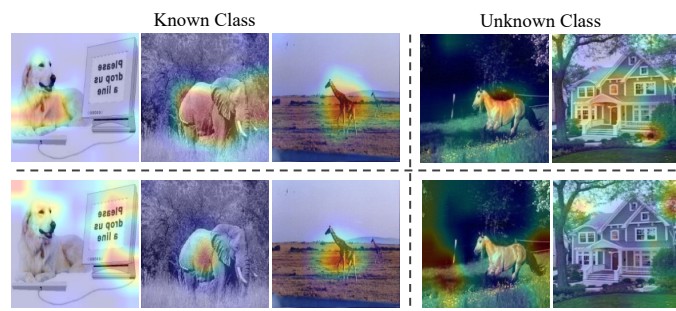

(a) Varying number of condensed samples.

(b) GradCAM visualization.

Figure 4: (a) Impact of varying number of condensed images per class of PACS. (b) GradCAM visualization of `ADA` (top) and ERM (bottom) on PACS.

an exploratory illustration. It shows that `ADA` demonstrates a tendency to focus on semantically meaningful regions. The activation maps generated by `ADA` are more comprehensive and precise, highlighting its ability to localize and understand objects in the image.

Figure 5 shows the UMAP (McInnes et al., 2018) visualization of features extracted by models trained with different methods: ERM, $UASC$ (the first stage of `ADA`), and the complete `ADA`. It reveals that ERM is deficient in discerning known and unseen classes, resulting in overlapping boundaries. $UASC$ improves this separation, and the full `ADA` yields the most distinct class boundaries.

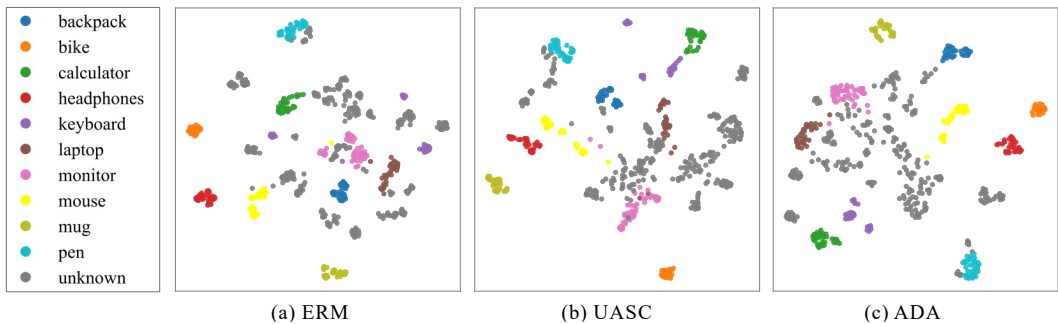

(a) ERM          (b) UASC          (c) ADA

Figure 5: UMAP feature visualization of different methods. Distinct colors refer to different categories while grey dots represent the unknown class data in the target domain.

# 5 Conclusion

In this paper, we propose `ADA` to tackle the OSMA problem, where both domain shift and open class challenges arise in the target domain. In the training stage, we uncover unknown class information hidden within the rich semantics of source domain known class images to enable the model to identify unknown class data. We additionally condense a small dataset to retain source domain data information, facilitating the subsequent adaptation process. During the test phase, we leverage the condensed data and target data to adapt the source-trained model to accommodate the open-set target domain. Empirically, `ADA` achieves superior performance on standard benchmarks in both online and offline settings.

# Acknowledgement

This work is supported by the Research Grants Council (RGC) of Hong Kong, SAR, China (GRF-16308321), and the NSFC/RGC Joint Research Scheme Grant N_HKUST635/20.

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

# A    Method Details

## A.1    Algorithm

Our proposed `Activate and Adapt (ADA)` consists of two stages: *Unknown Activation and Source Condensation (UASC)* and *Open Model Adaptation (OMA)*. Two stages are summarized in Algorithm 1.

---

**Algorithm 1** `Activate and Adapt (ADA)`

---

**Input**: Source domain data $\mathcal{D}_s = \{(x_i^s, y_i^s)\}_{i=1}^{N_s}$, feature extractor $f$, classifier $g$, domain classifiers $g_k$, $g_u$
**Output**: Label predictions for target domain data $\mathcal{D}_t = \{x_i^t\}_{i=1}^{N_t}$

1: STAGE 1: *UASC*
2: **for** source training epoch **do**
3:     Train $f$ and $g$ with $\mathcal{L}_{\mathrm{UA}}$ in Eq. 4
4: **end for**
5: Initialize condensed data $\mathcal{D}_{con} = \{(\tilde{x}_i, c)\}_{i=1}^{N_c}$ for each class $c \in \mathcal{C}_s$; freeze $f$
6: **for** condense training epoch **do**
7:     Optimize $\mathcal{D}_{con}$ with $\mathcal{L}_{\mathrm{SC}}$ in Eq. 5
8: **end for**
9: STAGE 2: *OMA*
10: **for** $x^t$ in $\mathcal{D}_t$ **do**
11:     Sample $(\tilde{x}, \tilde{y})$ from $\mathcal{D}_{con}$, and get a copy $(\tilde{x}', \tilde{y}')$
12:     Forward $x^t$ and $(\tilde{x}, \tilde{y})$ to get low-level features $\mathbf{z}_l$ and $\tilde{\mathbf{z}}_l$; inject target data style to $\tilde{\mathbf{z}}_l$ by Eq. 6 and then calculate $\mathcal{L}_{\mathrm{SI}}$ by Eq. 8
13:     Forward $x^t$ and $(\tilde{x}', \tilde{y}')$ to get $\mathcal{L}_{\mathrm{DA}}$ by Eq. 10
14:     Get total loss by Eq. 11 and backward propagate to update $f$, $g$, $g_k$, and $g_u$
15:     **if** Online **then**
16:         Predict for $x^t$
17:     **end if**
18: **end for**
19: **if** Offline **then**
20:     Predict for all $x^t$ in $\mathcal{D}_t$
21: **end if**

---

## A.2    Hyperparameters

**UASC**. We have detailed the source training parameters in the main paper. Here, we provide the parameters for the *Source Condensation* part. Since we adopt the ResNet-50 as the backbone, we apply the Instance Normalization to the output of the second residual block of ResNet-50. We adopt the SGD optimizer with a learning rate of 5.0, and we optimize the condensed images for 10,000 epochs. For each known class, we condense 10 images.

**OMA**. For the Style Injection operation, we apply it to the output of the second residual block of ResNet-50, aligning with the layer selected in the Instance Normalization in *UASC*.

## A.3    Model Selection

In the training stage, we train the model for 30 epochs for all datasets. We select the final epoch model for the subsequent *OMA* stage. It is justified in domain generalization, where the access to a train-validation split for model selection is typically unattainable.

## A.4    Baselines

Some baseline methods such as Tent (Wang et al., 2020) and ERM (Vapnik, 1999) are not specifically designed for handling the open-set classification scenario. We adapt these methods for the open-set classification

setting by predefining an entropy threshold (Zhu & Li, 2022). We set the threshold as $0.5 * \log |\mathcal{C}_s|$, where $|\mathcal{C}_s|$ is the number of known classes. Specifically, during the test phase, we calculate the entropy of the probability distribution for each data with respect to the $|C_s|$ dimensions. If the entropy is below the threshold, we predict it as known-class data, and the predicted class is the known class with the highest predicted probability. If the entropy is above the threshold, we predict it as an unknown class data.

## B More Results

### B.1 Additional Studies

We provide more results in this section.

**Impact of Test Order.** In our online setting, the target data arrives continuously, and for each batch of target data, the model updates on it and predicts presently. Here we investigate the order in which the target domain data arrives. For each order, we randomly shift the order of the whole target dataset. The result is present in Table 6. Our results indicate that the *OMA* phase exhibits robustness against variations in the order of target data, thereby evidencing its adaptability and robustness in open-world context.

Table 6: Impacts of the order of target data in the online setting.

| Order | OFFICE-HOME | OFFICE-31 | PACS | Average |
|-------|-------------|-----------|------|---------|
| 1 | 67.2 | 81.7 | 53.6 | 67.5 |
| 2 | 67.7 | 81.2 | 53.9 | 67.6 |
| 3 | 67.1 | 82.3 | 54.0 | 67.8 |
| 4 | 66.6 | 81.8 | 53.1 | 67.2 |

**Impact of Number of Known Classes.** We investigate the impact of the varying number of known classes on three methods including SHOT (Liang et al., 2020), AaD (Yang et al., 2022b), and our proposed method `ADA`. We keep the number of total classes ($|\mathcal{C}_s|+|\mathcal{C}_t^u|$) unchanged while changing the number of known class $|\mathcal{C}_s|$ in the source domain. As shown in Figure 6, our method consistently outperforms SHOT and AaD across settings with different numbers of known classes. Notably, the advantage of `ADA` is pronounced in scenarios with very few known classes. This highlights its effectiveness in situations where the supervisory information is scarce.

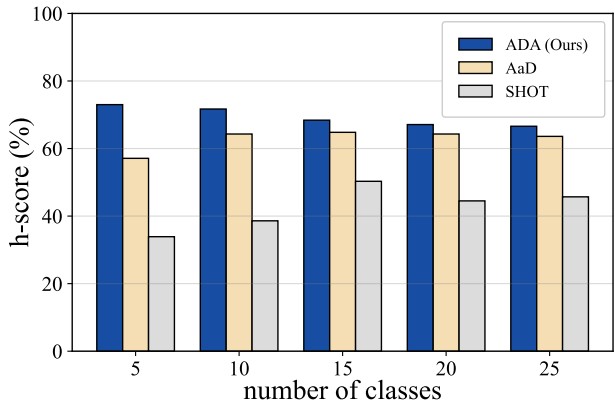

Figure 6: Impact of varying number of known classes for OFFICE-HOME.

**Study of Unknown Improvement Loss $\mathcal{L}_{\mathbf{UI}}$.** We investigate the $\mathcal{L}_{\mathrm{UI}}$ (Eq. 3) in two aspects. Firstly, we study the impact of the $L_2$ regularization term in $\mathcal{L}_{\mathrm{UI}}$, which serves to smooth the classifier's output, and subsequently to increase the probability of the unknown class. Table 7 presents the results ($hs$ after the first training stage) of using $L_2$ regularization and label smoothing. It shows that $L_2$ regularization performs

better than label smoothing across three datasets, while both these two methods surpass the case where no regularization term is included.

Table 7: $hs$ (%) of different methods in three benchmarks.

| Methods | OFFICE-HOME | OFFICE-31 | PACS |
|---|---|---|---|
| N/A | 63.4 | 77.1 | 50.4 |
| label smoothing | 64.1 | 78.3 | 50.8 |
| $L_2$ regularization | 66.0 | 80.1 | 51.8 |

Secondly, we study the impact of $\mathcal{L}_{\mathrm{UI}}$ on the known class performance $acc_k$. The unknown improvement loss $\mathcal{L}_{\mathrm{UI}}$ is designed to enhance the model's response to the unknown class in the absence of real unknown-class samples. As shown in the ablation results in Table 8, while $\mathcal{L}_{\mathrm{UI}}$ causes a slight decrease in $acc_k$, it significantly improves $acc_u$, resulting in a higher $hs$.

Table 8: Results (average) of different training losses on PACS.

| Losses | $acc_k$ | $acc_u$ | $hs$ |
|---|---|---|---|
| $\mathcal{L}_{\mathrm{UE}}$ | 54.6 | 45.9 | 48.6 |
| $\mathcal{L}_{\mathrm{UE}} + \mathcal{L}_{\mathrm{UI}}$ | 50.9 | 55.7 | 51.8 |

**Results on Higher Resolution Backbone.** We provide results of `ADA` and other methods on higher resolution backbone (ResNet-101 with $384 \times 384$ input size). Note that our method applies to CNN architectures since it relies on spatially local features (Section 3.1.1). Results in Table 9 show that `ADA` outperforms baselines with this larger backbone, and they also surpass those in our main results (ResNet-50).

Table 9: Results with ResNet-101 backbone.

| Methods | OFFICE-HOME | | | OFFICE-31 | | | PACS | | |
|---|---|---|---|---|---|---|---|---|---|
| | $acc_k$ | $acc_u$ | $hs$ | $acc_k$ | $acc_u$ | $hs$ | $acc_k$ | $acc_u$ | $hs$ |
| Tent | **77.1** | 50.9 | 60.2 | **85.4** | 61.0 | 69.1 | 41.4 | 53.3 | 45.9 |
| OSTTA | 63.2 | 70.1 | 65.4 | 79.2 | 79.2 | 78.8 | 57.2 | 50.4 | 53.5 |
| UniEnt | 70.2 | 62.4 | 66.3 | 79.2 | 82.0 | 80.3 | **67.9** | 47.7 | 53.3 |
| ART | 64.3 | 72.0 | 67.9 | 82.3 | 80.4 | 81.3 | 62.1 | 46.3 | 53.8 |
| `ADA` (Online) | 64.5 | **76.1** | **69.1** | 80.5 | **84.8** | **82.1** | 60.7 | **54.5** | **55.9** |
| ERM | 57.1 | 55.5 | 55.8 | **92.2** | 65.2 | 72.9 | 31.7 | **66.6** | 42.0 |
| SHOT | **72.1** | 55.4 | 60.0 | 78.6 | 55.7 | 65.8 | **68.9** | 42.3 | 49.1 |
| AaD | 67.8 | 72.7 | 68.1 | 79.7 | 80.4 | 80.5 | 67.8 | 53.0 | 57.5 |
| OneRing | 64.7 | 73.9 | 67.6 | 79.8 | **87.7** | 81.3 | 62.4 | 50.3 | 55.9 |
| `ADA` (Offline) | 66.3 | **76.6** | **70.4** | 81.5 | 85.4 | **83.0** | 61.9 | 55.3 | **59.2** |

**Results on More Datasets.** We additionally provide results on VisDA (Peng et al., 2017). Six classes are designated as known classes $\mathcal{C}_s$ (bicycle, bus, car, motorcycle, train, truck), and the remaining six classes are treated as the unknown class $\mathcal{C}_t^u$. Results in Table 10 show that `ADA` consistently outperforms other methods in both offline and online settings.

**Comparison with Other Baselines.** We compare our method with more methods, including Novel Class Discovery (ComEx (Yang et al., 2022a), IIC (Li et al., 2023a)), inheritable model (Kundu et al., 2020), SFDA² (Hwang et al., 2024), and CODA (Chen et al., 2023a). Results in Table 11 show that `ADA` outperforms these methods on three benchmarks.

Table 10: Results on VisDA.

| Methods | $acc_k$ | $acc_u$ | $hs$ |
|---|---|---|---|
| Tent | **55.3** | 40.2 | 46.6 |
| OSTTA | 46.8 | 72.0 | 56.7 |
| UniEnt | 52.3 | 70.1 | 59.9 |
| ART | 51.3 | **77.2** | 61.6 |
| ADA (Online) | 52.9 | 77.1 | **62.7** |
| ERM | 61.2 | 39.9 | 48.3 |
| SHOT | 51.5 | 59.8 | 55.3 |
| AaD | 48.0 | **87.3** | 61.9 |
| OneRing | **62.1** | 59.5 | 60.8 |
| ADA (Offline) | 53.0 | 80.1 | **63.8** |

Table 11: Comparison with more methods.

| Methods | OFFICE-HOME | | | OFFICE-31 | | | PACS | | |
|---|---|---|---|---|---|---|---|---|---|
| | $acc_k$ | $acc_u$ | $hs$ | $acc_k$ | $acc_u$ | $hs$ | $acc_k$ | $acc_u$ | $hs$ |
| ComEx | 57.8 | 63.9 | 57.1 | 77.6 | 68.6 | 72.6 | 50.8 | 59.8 | 53.0 |
| IIC | 57.4 | 71.3 | 61.7 | 76.6 | 75.3 | 75.8 | 48.6 | 56.0 | 49.4 |
| Inheritable | **67.6** | 65.0 | 65.3 | 84.4 | 77.0 | 80.2 | 51.5 | 59.8 | 54.0 |
| SF(DA)$^2$ | 64.7 | 69.0 | 65.7 | **85.0** | 74.5 | 79.3 | **64.0** | 49.6 | 55.1 |
| CODA | 62.4 | 73.3 | 66.7 | 83.7 | 78.2 | 80.5 | 53.3 | 64.4 | 57.6 |
| ADA (Offline) | 63.1 | **76.7** | **68.4** | 84.5 | **81.3** | **82.7** | 53.9 | **71.7** | **58.7** |

## B.2   GradCAM Visualization

The additional GradCAM visualizations (Selvaraju et al., 2017) of ADA and ERM are shown in Figure 7. Consistent with the result in the main paper, it can be observed that the hot zones activated by ADA are more accurate and comprehensive compared to ERM. Hence, ADA provides a more thorough understanding of the images in the open-set and distribution-shifted context.

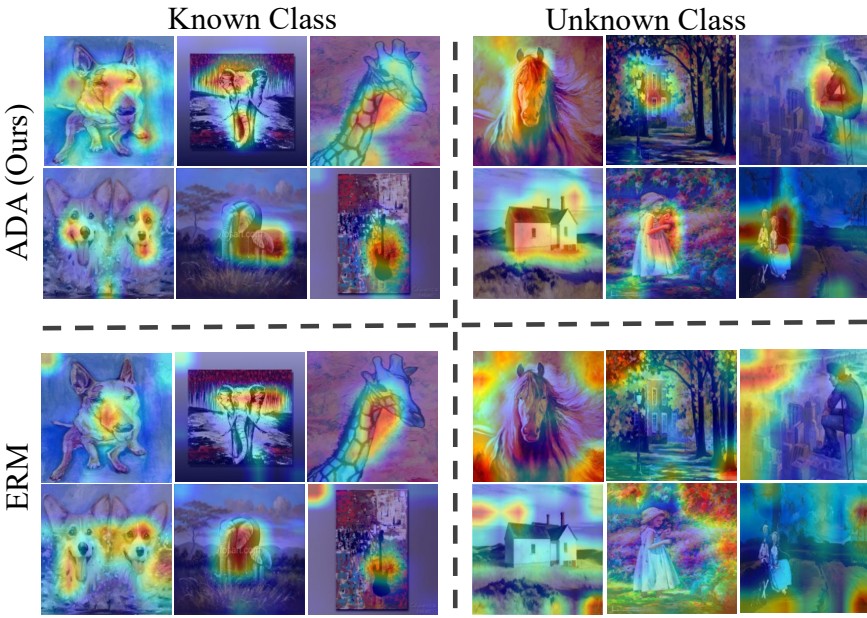

Figure 7: GradCAM visualization of ADA and ERM.

### B.3 Visualization of Condensed Dataset

Figure 8 shows the condensed images of three datasets. We can see that these images look like noise images and do not show any resemblance to the original training data visually. Thus, it prevents information leakage of the source domain training data to some extent. Moreover, as elaborated in our main paper, it preserves information that facilitates our model adaptation process.

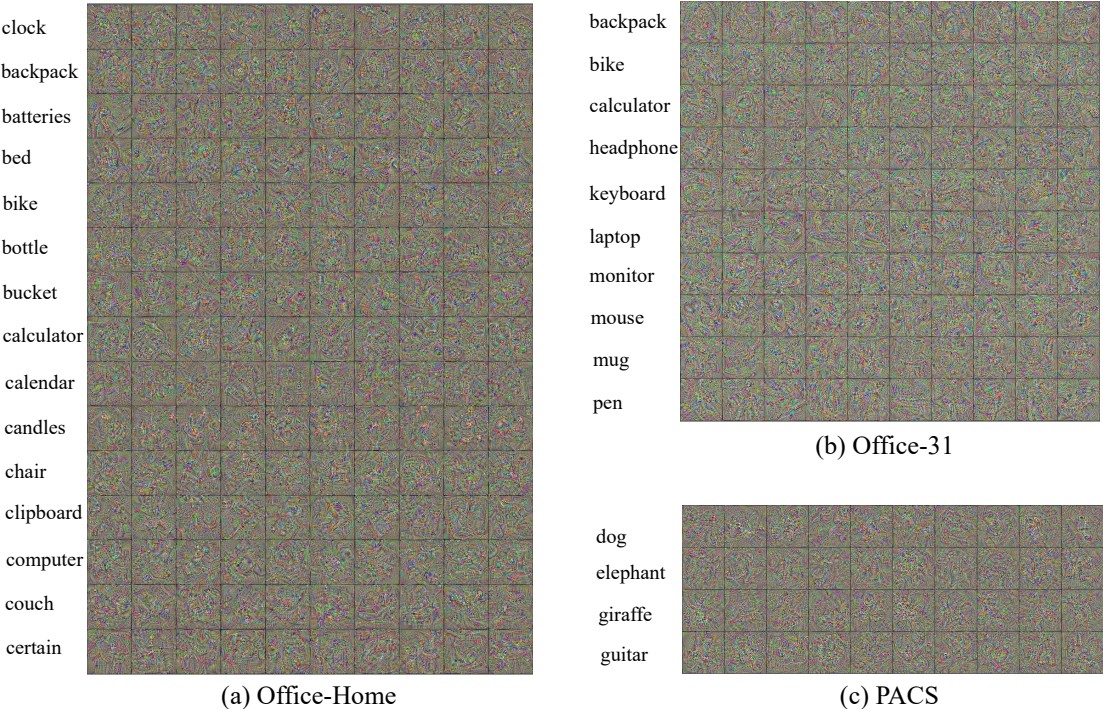

Figure 8: Visualization of condensed data of OFFICE-HOME (Artistic), OFFICE-31 (Amazon), and PACS (Photo). Each row shows 10 images per (known) class.

## C Limitations

Our approach has limitations. Firstly, `ADA` treats all open classes as a single category (the unknown class $(|\mathcal{C}_s|+1)$-th dimension in the classifier), which restricts its broader applicability. In situations that require the identification of individual open classes, `ADA`'s effectiveness is not assured. Secondly, `ADA` relies on condensing the source domain data to a smaller dataset. Therefore, the success of `ADA` depends on the advancements in dataset condensation techniques. As the number of images and categories increases, the complexity of the dataset condensation process also rises. However, we anticipate that improvements in dataset condensation techniques, along with advances in deep learning hardware, will help address this challenge.

