# OpenReview forum: "Activate and Adapt: A Two-Stage Framework for Open-Set Model Adaptation"
_TMLR — Accepted by TMLR_

### Review · Reviewer_nB99 · 2025-04-12

**Summary Of Contributions:**

The paper presents a generalization of the test time adaptation problem. The test domain can undergo both domain shifts and additional classes during test time in their version. To handle this, they utilize the semantic information in the source classes to find the new classes in the test domain. At the same time, they also condense the information in the source domain to use it as a remembering mechanism during adaptation.

**Audience:**

Yes

**Broader Impact Concerns:**

There are no adverse broader impact of the work.

**Claims And Evidence:**

Yes

**Requested Changes:**

1. Additional experiments with Novel class discovery methods such as [1] and other modern baselines.
2. Instead of using condensed data, can you subsample from the source dataset and use that as your condensed dataset and then report the results? This will help validate the utility of data condensation.
3. If the above is done, then use it to do OMA. After that, it will effectively serve as doing OMA without UASC, and I believe it will add proof to the independent usefulness of the components.

[1] Vaze, Sagar, et al. "Generalized category discovery." Proceedings of the IEEE/CVF conference on computer vision and pattern recognition. 2022.

**Strengths And Weaknesses:**

Strengths:
1. The paper is well-written and timely. Stable test time training is an important problem statement, especially in the age of large models with even longer training times.
2. The motivations of the proposed components are intuitive and reasonable.
3. The ablation study in Table 4 is extensive.
4. Limitations are discussed in the Appendix

Weakness:
1. The experimental section has supporting evidence that the methods are working as expected and better than existing methods. Your method handles two things domain shit and novel class addition. I think comparing it with existing novel class discovery methods will also be useful in showing the relative performance of your method, even if the train and test protocol are not completely identical.

---

> ### Author Response · Authors · 2025-05-22
> **Response to Reviewer nB99**
>
> Thank you for your effort in reviewing our paper and for the insightful comments. We address your concerns as follows, and we incorporate your suggestions into our revised manuscript.
>
> ---
> > ​**Weakness 1 & Change 1: Compare with Novel Category Discovery methods.**
>
> Thanks for your advice. Since the paper you mentioned applies to ViT, while our method applies to CNN as it relies on spatially local features, we compare our method with NCD methods that apply to CNN architectures, including ComEx [1] and IIC [2]. The results are as follows. It shows that our method surpasses these NCD methods across three benchmarks. It can be attributed to the fact that NCD methods primarily address the open class problem. However, in our setting, we face dual challenges of both domain shift and open class, which is more challenging.
>
> | Methods    | OfficeHome         | Office-31          | PACS                |
> |------------|--------------------|--------------------|---------------------|
> | ComEx [1]  | 57.8 / 53.9 / 57.1 | 77.6 / 68.6 / 72.6 | 50.8 / 59.8 / 53.0  |
> | IIC [2]    | 57.4 / 71.3 / 61.7 | 76.6 / 75.3 / 75.8 | 48.6 / 56.0 / 49.4  |
> | ADA (Ours) | 63.1 / 76.7 / **68.4** | 84.5 / 81.3 / **82.7** | 53.9 / 71.7 / **58.7**  |
> (metrics: $acc_k$ / $acc_u$ / $hs$, $hs$ is the key metric)
> ---
>
> > ​**Change 2 & Change 3: Validate the utility of data condensation. For example, subsample from the source dataset and use it as the condensed dataset.**
>
> Thanks for your advice. In our original manuscript (Table 5), we analyze the benefits of employing a condensed dataset over the raw source domain data. Specifically, we evaluate our method against two alternatives:  (1) randomly sampling a subset from source training data; (2) selecting samples closest to each class’s prototype.
>
> For your convenience, we present the results in the paper ($hs$ on PACS) as below. It shows that our strategy outperforms both alternatives. Moreover, utilizing source domain data carries the risk of data leakage, whereas using a condensed dataset avoids this issue.
>
> | Strategies  | Random Selection | Class Prototype | Condensed Dataset |
> |-------------|------------------|-----------------|-------------------|
> | h-score (%) | 56.9             | 56.7            | 58.7              |
>
> ---
> References
>
> [1] M. Yang et al., 2022, “Divide and conquer: compositional experts for generalized novel class discovery”, CVPR.
>
> [2] W. Li, et al., 2023, “Modeling Inter-Class and Intra-Class Constraints in Novel Class Discovery”, CVPR.

---

### Review · Reviewer_xmJo · 2025-04-14

**Summary Of Contributions:**

This paper addresses the challenging problem of Open-Set Adaptation, where a model trained on source domain data containing only known classes needs to be adapted to a target domain with distribution shift and new unknown classes. The authors propose a novel two-stage framework called Activate and Adapt.
In the first stage, the model extracts unknown class information from the rich semantics within the source domain data and condenses the source dataset while preserving domain information.
In the second stage, the model is adapted to the target domain by injecting the target domain style into the condensed dataset and decoupling domain alignment for known and unknown classes.
The framework is evaluated on three standard benchmarks: Office-Home, Office-31, and PACS, in both online and offline settings.

**Audience:**

Yes

**Claims And Evidence:**

No

**Requested Changes:**

1. Please experiment with higher-resolution backbones (e.g., ViT).
2. Compare your fine-grained method with the one from inheritable models for open-set domain adaptation.
3. Quantify iteration costs in online settings between your method and others.
4. Please see weaknesses above

**Strengths And Weaknesses:**

**Strengths**

1. The paper addresses an important practical problem that combines domain shift with open-set challenges
2. The two-stage approach offers a novel solution that doesn't require predefining thresholds for unknown detection
3. The experimental evaluation is comprehensive across multiple datasets and settings (online/offline)
4. The method outperforms previous approaches in both settings on all benchmarks

**Weaknesses**

1. The explanation of how unknown classes are extracted from source domain data is unclear. The paper claims to utilize "rich semantics" in source domain data, but the mechanism by which misclassified granular features represent unknown classes needs stronger justification. How do we know these misclassifications correspond to potential unknown categories rather than just difficult examples of known classes, and at what points in the training do we define them as unknown?

2. The decomposition of feature maps into "granular features" (Equation 1) may be problematic. For ResNet-50, the penultimate layer typically has 7×7 spatial resolution, meaning each granular feature corresponds to a 32×32 pixel region in the original image (assuming 224×224 inputs).

3. The statement "Source domain data is often unavailable during adaptation due to safety and data privacy concerns. Most previous methods simply discard it during adaptation" is confusingly written. It implies methods discard data for safety/privacy reasons, but in reality, source-free methods are designed specifically to operate without source data because of these constraints.

4. Please reformulate  "after forwarding the fixed feature extractor f."

5. The definition of "upper and lower parts" of the feature extractor is ambiguous: "We denote f = f1 ◦ f2, where f1 and f2 denote the upper and lower parts of f, respectively." Which layers specifically constitute f1 versus f2?

6. The Source Condensation component introduces additional computational overhead and complexity. Is this condensation step truly necessary? The paper claims it addresses privacy concerns, but creating and transferring condensed data still requires additional compute and storage resources. Could the method be simplified to work directly with the trained source model without this intermediate representation? One may claim that the condensed dataset can be reconstructed using inversion methods.

7. The unknown improvement loss in Equation 3 uses L2-norm regularization without sufficient justification over simpler alternatives like label smoothing.

8. In Tables 2-3, the online h-score is surprisingly close to the offline h-score on Office-Home, for example. Could you elaborate on why this is the case?

9. The visualization in Figure 4(b) compares ADA's GradCAM with ERM, but ERM lacks unknown-class training, making this comparison potentially misleading. How do you actually do the ERM GradCAM if you don't have the unknown class?

10. The related work section should include recent advances in source-free open-set domain adaptation, such as GLC, Lead, as well as a discussion of inheritable models for open-set domain adaptation.

---

> ### Author Response · Authors · 2025-05-22
> **Response to Reviewer xmJo (1/2)**
>
> Thank you for your effort in reviewing our paper and for your constructive feedback. We address your concerns as follows, and we incorporate your suggestions into our revised manuscript.
>
> ---
> > ​**Weakness 1: The claim that misclassified granular features represent the unknown class needs justification. At what points do we define them as unknown?**
>
> Thanks for the insightful question. We assign “pseudo unknown-class labels” to the granular features whose predicted labels disagree with the image label, and utilize them to activate the unknown-class logit during training. Note that our goal is **not to ensure precise discovery of unknown-class granular features, but to endow the model with the ability to identify unknown-class data**. This process starts from the beginning of training. As Eq. 4 illustrates, the cross-entropy loss guarantees the known-class performance, while other terms enhance the model's ability to identify unknown-class data.
>
> ---
> > **Weakness 2: Explain the decomposition of feature maps into granular features. Each granular feature corresponds to a region in the original image.**
>
> Thanks for your question. Your understanding is correct that each unit ($C\times1\times1$) in the penultimate-layer feature map (C$\times$H$\times$W) corresponds to a localized region in the input image. Our approach is motivated by discovering potential new-class regions (rather than new-class pixels) embedded within the known-class images.
>
> ---
> > **Weakness 3: Explain "Source domain data is often unavailable during adaptation, and most methods simply discard it". Source-free methods are designed to operate without source data.**
>
> Thanks for your question. To clarify our motivation more clearly, we rephrase it in our revised version as follows.
>
> Due to safety and data privacy concerns, source-free domain adaptation and test-time adaptation methods arise to adapt a trained model to the target domain without accessing source training data. However, discarding source data entirely during adaptation can restrict potential performance gains, as it eliminates the only source of precise supervision. This motivates us to leverage source data while ensuring privacy. We achieve this by compressing the source data into a condensed dataset that retains class-wise information.
>
> ---
> > **Weakness 4: Rephrase "match their feature distributions after forwarding the fixed feature extractor f".**
>
> Thanks for your advice. We have rephrased it as "minimize the $L_2$ distance between the means of their features extracted by the fixed feature extractor $f$". We also rephrased our source condensation section in the revised version.
>
> ---
> > **Weakness 5: Specify which layers constitute $f_1$ and $f_2$ in $f=f_1\circ f_2$.**
>
> Thanks for pointing this out. In our experiments, we adopt ResNet-50 ($f$) as the backbone, which consists of four residual blocks. In $f=f_1\circ f_2$, $f_2$ comprises the first and second residual blocks of $f$, representing the lower part of $f$, while $f_1$ comprises the third and fourth residual blocks, representing the upper part of $f$. We apply instance normalization (in Source Condensation) and style injection to the output of the $f_2$, as previous works [1][2] have shown that the statistics of the features after these blocks describe the style of images.
>
> ---
> > **Weakness 6: Is the Source Condensation module necessary?**
>
> Thanks for your question. We demonstarte the necessity of the Source Condensation module in two aspects: (1) when SC (and consequently, the second model adaptation stage) is ablated, the performance degrades (Table 4); (2) when the source condensed dataset is replaced by alternative strategies, including (i) randomly sampling a subset from source training data, and (ii) selecting samples closest to each class’s prototype, the performance degrades (Table 5).
>
> The results are shown in Tables 4 and 5 in the original manuscript. For your convenience, we present the results in the paper ($hs$ on PACS) as follows.
>
> |Methods|Ablate SC|Random Selection|Class Prototype|Condensed Dataset|
> |-|-|-|-|-|
> | h-score (%) |51.8|56.9|56.7|58.7|
>
> ---
> > **Weakness 7: Eq. 3 uses $L_2$ regularization loss without sufficient justification over simple alternatives like label smoothing.**
>
> Thanks for your advice. The $L_2$ regularization in Eq. 3 serves to smooth the classifier’s output, and subsequently to increase the probability of the unknown class. The following table presents the results ($hs$ after the first training stage) of using $L_2$ regularization and label smoothing. It shows that $L_2$ regularization performs better than label smoothing across three datasets, while both these two methods surpass the case where no regularization term is included.
>
> | Methods|Office-Home|Office-31|PACS|
> |-|-|-|-|
> | N/A| 63.4 | 77.1| 50.4 |
> | label smoothing| 64.1| 78.3| 50.8 |
> | $L_2$ regularization | 66.0 | 80.1| 51.8 |
>
> ---
> (continue)

---

> ### Author Response · Authors · 2025-05-22
> **Response to Reviewer xmJo (2/2)**
>
> > **Weakness 8: Explain h-score of the online setting is close to that of the offline setting for Office-Home.**
>
> Thanks for your question. As shown in our results, the h-scores under online and offline settings are closer on Office-Home and Office-31 than on PACS. This can be explained by the relative inter-domain discrepancies for different datasets. Specifically, **Office-Home and Office-31 exhibit smaller inter-domain discrepancies compared to PACS**. For example, in PACS, sketch images only show the simple contours compared to photo images, leading to large domain discrepancies. The smaller domain gap (in Office-Home and Office-31) reduces the difficulty of adaptation in the online setting and thus narrows the performance gap with the offline setting.
>
> ---
> > **Weakness 9: Explain how to visualize the GradCAM for ERM since ERM lacks unknown-class training.**
>
> Thanks for your question. While ERM is not trained with unknown-class supervision, we include the $(C_s+1)$-th output node as a structural placeholder in the classifier. For ERM, the unknown-class identification is based on entropy thresholding rather than on this output node (detailed in Appendix A.4). For visualization, we apply GradCAM to the $(C_s+1)$-th output to heuristically examine how ERM responds to the real unknown-class inputs. We agree that this comparison is not strictly rigorous, and we clarify its exploratory nature and interpretive limitations in the revised version.
>
> ---
> > **Weakness 10: Related work should include source-free OSDA and inheritable models.**
>
> Thanks for your advice. We have added the related works in the revised version.
>
> ---
> > **Change 1: Experiment with higher resolution backbone.**
>
> Thanks for your advice. Our method applies to CNN architectures since it relies on spatially local features. To address your concern, we adopt a higher-resolution CNN architecture, ResNet101 with $384\times384$ input size, as the backbone to evaluate the effectiveness of our method. The results show that our method outperforms other baselines with this new backbone, and the results also surpass those in our original manuscript (ResNet-50).
>
> | Methods | OfficeHome | Office-31| PACS |
> |-|-|-|-|
> | Tent          | 77.1 / 50.9 / 60.2   | 85.4 / 61.0 / 69.1 | 41.4 / 53.3 / 45.9 |
> | OSTTA         | 63.2 / 70.1 / 65.4   | 79.2 / 79.2 / 78.8 | 57.2 / 50.4 / 53.5 |
> | UniEnt        | 70.2 / 62.4 / 66.3   | 79.2 / 82.0 / 80.3 | 67.9 / 47.7 / 53.3 |
> | ART           | 64.3 / 72.0 / 67.9   | 82.3 / 80.4 / 81.3 | 62.1 / 46.3 / 53.8 |
> | ADA (Online)  | 64.5  / 76.1 / **69.1**  | 80.5 / 84.8 / **82.1** | 60.7 / 54.5 / **55.9** |
> | ERM           | 57.1 / 55.5 / 55.8   | 92.2 / 65.2 / 72.9 | 31.7 / 66.6 / 42.0 |
> | SHOT          | 72.1 / 55.4 / 60.0   | 78.6 / 55.7 / 65.8 | 68.9 / 42.3 / 49.1 |
> | AaD           | 67.8 / 72.7 / 68.1   | 79.7 / 80.4 / 80.5 | 67.8 / 53.0 / 57.5 |
> | OneRing       | 64.7 / 73.9 / 67.6   | 79.8 / 87.7 / 81.3 | 62.4 / 50.3 / 55.9 |
> | ADA (Offline) | 66.3 / 76.6 / **70.4**   | 81.5 / 85.4 / **83.0** | 61.9 / 55.3 / **59.2** |
> (metrics: $acc_k$/$acc_u$/$hs$, $hs$ is the key metric)
>
> ---
> > **Change 2: Compare with the inheritable method in OSDA.**
>
> | Methods  | OfficeHome | Office-31 | PACS |
> |-|-|-|-|
> | Inheritable [3] | 67.6 / 65.0 / 65.3 | 84.4 / 77.0 / 80.2 | 51.5 / 59.8 / 54.0 |
> | ADA (Ours)      | 63.1 / 76.7 / **68.4** | 84.5 / 81.3 / **82.7** | 53.9 / 71.7 / **58.7** |
> (metrics: $acc_k$/$acc_u$/$hs$, $hs$ is the key metric)
>
> ---
> > **Change 3: Quantify iteration costs in online setting between your method and others.**
>
> Thanks for your question. For all methods in the online setting, each target domain data passes through the model only once. More specifically, when a batch of target data arrives, the model adapts on it and makes predictions presently. There is no inner loop for updating on the batch of data. Hence, the iteration costs of **all methods are the same (1 iteration)** in the online setting.
>
> ---
> References
>
> [1] K. Zhou, et al., 2021, “Domain generalization with mixstyle”, ICLR.
>
> [2] J. Kang, et al., 2022, “Style neophile: constantly seeking novel styles for domain generalization”, CVPR.
>
> [3] J.N. Kundu. et al., 2020, “Towards Inheritable Models for Open-Set Domain Adaptation”, CVPR.

---

> > ### Comment · Reviewer_xmJo · 2025-06-10
> >
> > Thank you for your detailed response to my comments. The additional experiments and the revised wording have strengthened the manuscript, and I am satisfied with the changes.

---

### Review · Reviewer_Scfk · 2025-05-10

**Summary Of Contributions:**

The paper proposes a novel technique for open-set model adaptation. In this setting, a model is first trained on the source domain and then adapted to the target domain which can also have certain new classes. The authors propose a two-stage framework: ADA, Activate and Adapt for effective model adaptation. They extract new class information from the source domain data and use it for training the model to be able to identify unknown samples. To prevent forgetting during adaptation, they condense the source domain data into a small dataset. Furthermore, they adapt the model to the target domain at test time by injecting target style information into the condensed data providing supervision for effective recognition of both known and unknown target classes. Moreover, a domain alignment loss is introduced to make the model insensitive to the domain variances.

**Audience:**

Yes

**Claims And Evidence:**

Yes

**Requested Changes:**

1. It’s not clear how the authors get the condensed dataset. It would be nice if they can add further details in that subsection.

2. The methods used for comparison are not the most recent ones. The authors should consider adding comparisons with some more recent approaches like:

Hwang, U., Lee, J., Shin, J. and Yoon, S., 2024. SF (DA) $^ 2$: Source-free domain adaptation through the lens of data augmentation. arXiv preprint arXiv:2403.10834.

Kundu, J.N., Kulkarni, A.R., Bhambri, S., Mehta, D., Kulkarni, S.A., Jampani, V. and Radhakrishnan, V.B., 2022, June. Balancing discriminability and transferability for source-free domain adaptation. In International conference on machine learning (pp. 11710-11728). PMLR.

3. DA papers usually also experiment on VisDA and DomainNet datasets. Can the authors also add experiments on some of these datasets in the paper?
4. It would be nice to also have an ablation to see how the unknown improvement loss affects the known class accuracy.

**Strengths And Weaknesses:**

Strengths:

1. The authors’ tackle the problem of open-set test time adaptation which is quite practical but underexplored.

2. The idea of extracting features of new classes from the source domain images and using it for training the model to be able to identify unknown samples is interesting.

3. The style injection from the target domain into the condensed data helps in providing supervision for adapting the model at test-time.

4. The domain alignment loss further makes the model insensitive to the domain variance.

Weaknesses:

1. The authors’ use a domain alignment loss where for the granular features from condensed data the probability that they belong to the target data is maximized. If the binary classifiers gk and gu are trained with this loss, won’t they just learn a reverse mapping? I am not sure if I am missing something here.

2. The accuracy of the proposed approach is lower than other methods for known class identification. Is this because of the unknown improvement loss which increases the probability of the sample being unknown even for known class samples? Can the authors explain why the known class performance is not good enough?

---

> ### Author Response · Authors · 2025-05-22
> **Response to Reviewer Scfk**
>
> Thank you for your efforts in reviewing our paper, and valuable suggestions. We address your concerns as follows, and we incorporate your suggestions into our revised manuscript.
>
> ---
> > **Weakness 1: Explain domain alignment loss. For granular features from condensed data, the probability that they belong to the target domain is maximized. Won’t this lead $g_k$ and $g_u$ to learn a reverse mapping?**
>
> Thanks for your question. We design the domain alignment loss to encourage the model to be **insensitive** to domain variations. For example, for granular features from condensed data, we adapt the model with the loss that maximizes the predicted probability of belonging to the target domain. In this way, the model cannot tell which domain the feature is from, thereby raising the model’s insensitivity to domain variations and increasing its generalizability across different domains.
>
> ---
> > **Weakness 2 & Change 4: Explain why the known class performance $acc_k$ is not good enough. Is it because of the unknown improvement loss? Add an ablation study to see how unknown improvement loss affects known class accuracy.**
>
> Thanks for your insightful question and suggestion. In our problem setting, we emphasize the model’s performance on both known-class classification ($acc_k$) and unknown-class identification ($acc_u$). The hscore ($hs$) serves as the core metric, and a high $hs$ requires both $acc_k$ and $acc_u$ to be high and balanced. The unknown improvement loss $\mathcal{L}_{\text{UI}}$ is designed to enhance the model’s response to the unknown class in the absence of real unknown-class samples.
>
> As shown in the ablation results below, while $\mathcal{L}_{\text{UI}}$ causes a slight decrease in $acc_k$, it significantly improves $acc_u$, resulting in a higher $hs$.
>
> | Losses | $acc_k$ | $acc_u$ | $hs$ |
> |-|-|-|-|
> | UE | 54.6    | 45.9    | 48.6 |
> | UE+UI | 50.9    | 55.7    | 51.8 |
>
> (Average results of all experiments on PACS)
>
> ---
> > **Change 1: Add further details in the source condensation section.**
>
> Thanks for your suggestion. We have added more details in the source condensation section (Section 3.1.2) in our revised version.
>
> ---
> > **Change 2: Consider adding comparisons with more methods.**
>
> Thanks for your advice. Since the code of the second paper you mentioned is not available, we compare our method with two related and more recent methods, including SF(DA)^2 [1] and CODA [2]. The results show that our method surpasses these two baselines on three benchmarks.
>
> | Methods | OfficeHome | Office-31| PACS  |
> |-|-|-|-|
> | SF(DA)^2 [1] | 64.7 / 69.0 / 65.7 | 85.0 / 74.5 / 79.3 | 64.0 / 50.0 / 55.1 |
> | CODA [2]     | 62.4 / 73.3 / 66.7 | 83.7 / 78.2 / 80.5 | 63.3 / 64.4 / 57.6 |
> | ADA (Ours)   | 63.1 / 76.7 / **68.4** | 84.5 / 81.3 / **82.7** | 53.9 / 71.7 / **58.7** |
> (metrics: $acc_k$/$acc_u$/$hs$, $hs$ is the key metric)
>
> ---
> > **Change 3: Consider adding experiments with VisDA and DomainNet.**
>
> Thanks for your advice. We add experiments with the VisDA dataset. The results are presented in the following table. It shows that ADA consistently outperforms other methods in both offline and online settings, consistent with our main findings.
>
> However, applying our method to the DomainNet presents practical challenges. We have claimed in the limitation section (Appendix C) that our method's source condensation becomes computationally intensive as the scale of the dataset grows. DomainNet's size (345 categories, 569,010 images) makes the condensation process difficult to implement efficiently.
>
> | Methods       | $acc_k$ / $acc_u$ / $hs$  |
> |--|--|
> | Tent          | 55.3 / 40.2 / 46.6        |
> | OSTTA         | 46.8 / 72.0 / 56.7        |
> | UniEnt        | 52.3 / 70.1 / 59.9        |
> | ART           | 51.3 / 77.2 / 61.6        |
> | ADA (Online)  | 52.9 / 77.1 / **62.7**        |
> | ERM           | 61.2 / 39.9 / 48.3        |
> | SHOT          | 51.5 / 59.8 / 55.3        |
> | AaD           | 48.0 / 87.3 / 61.9        |
> | OneRing       | 62.1 / 59.5 / 60.8        |
> | ADA (Offline) | 53.0 / 80.1 / **63.8**        |
>
> ---
> References
>
> [1] U. Hwang, et al., 2024, “SF(DA)^2: source free domain adaptation through the length of data augmentation”, ICLR.
>
> [2] C. Chen, et al., 2023, “CODA: Generalizing to Open and Unseen Domains with Compaction and Disambiguation”, NeurIPS.

---

### Author Response · Authors · 2025-05-22
**​​Authors' Response to Reviewers' Comments (Revised Manuscript Version)​​**

We sincerely thank all the reviewers for their time, insightful comments, and valuable suggestions. We respond to each reviewer’s comments in detail below, and we have incorporated all the reviewers' suggestions into our manuscript to strengthen our paper. All revisions are highlighted in blue in the updated manuscript. The main changes we made include:

---
**Additional Results**: (Appendix B.1)

- Extended study on the Unknown Improvement loss, including $L_2$ regularization (xmJo), and how UI loss impacts known class accuracy (Scfk)

- Higher resolution backbone results (xmJo)

- Comparison with additional methods, including NCD methods (nB99), inheritable model (xmJo), and more recent methods (Scfk)

- Additional dataset evaluation (Scfk)

**Methodological Clarifications**:

- Rephrase and elaborate Source Condensation Section (Section 3.1.2, xmJo, Scfk)

- Details of backbone (Appendix A.2, xmJo)

- Explanation of visualization (Section 4.4, xmJo)

**Related Work**: (Section 2)

- More source-free OSDA related works (xmJo)

---

### Decision · Action_Editor_SdU1 · 2025-06-25

**Recommendation:** Accept as is

**Additional Comments:**

Reviewer consensus is that the submission satisfies both the Claims and evidence and Audience criteria and therefore meets the bar for acceptance at TMLR.

**Audience:**

Yes

**Audience Explanation:**

All three reviewers find that the submission if of interest to at least part of TMLR's readership:

- "[The] problem of open-set test time adaptation [...] is quite practical but underexplored." (Scfk)
- "The paper addresses an important practical problem that combines domain shift with open-set challenges." (xmJo)
- "Stable test time training is an important problem statement, especially in the age of large models with even longer training times." (nB99)

**Claims And Evidence:**

Yes

**Claims Explanation:**

Reviewers' initial concerns over claims and evidence have been addressed to their satisfaction by the authors, and all three reviewers find that the claims made in the submission are supported by accurate, convincing and clear evidence:

- "The work is technically sound and satisfies claims made in the paper." (Scfk)
- "The experimental evaluation is comprehensive across multiple datasets and settings (online/offline)." (xmJo)
- "I am satisfied with the results and the additional experiments done by the authors." (nB99)